# The Quality-Utility Paradox: Why High-Reward Data Impairs Small Model Mathematical Reasoning

Haolong Qian [1]  Xianliang Yang [2]  Yinuo Ma [1]  Lirong Che [1]  Feng Lu [3]  Ye Guo [4]  Lei Song [2]  Jiang Bian [2]  Chun Yuan [1]

## Abstract

Knowledge distillation from powerful reasoning models is widely used to improve Small Language Models (SLMs) on mathematical reasoning, often assuming that traces with higher reward model scores provide more useful supervision. We identify a counterintuitive **Quality-Utility Paradox** in mathematical reasoning distillation. Data refined or synthesized by a stronger Oracle obtains higher perceived quality according to reward models, yet consistently underperforms traces generated by the SLM itself and selected through rejection sampling across Qwen2.5, LLaMA-3, and DeepSeek families. Our analysis shows that Oracle refinement couples logical repair with distributional drift away from the SLM's native reasoning distribution. This drift increases the learner's adaptation cost and can outweigh the benefit of improved reasoning logic. To test this mechanism, we introduce **Style-Aligned Refinement**, which preserves the native trajectory of the SLM while retaining logical repair from the Oracle. This intervention lowers adaptation cost and restores downstream utility. These findings suggest that effective mathematical reasoning distillation should jointly optimize perceived solution quality and learner-data compatibility, rather than relying solely on reward-model scores. The datasets and code are available at https://github.com/Dracoqhl/Quality-Utility-Paradox.

---

[1]Tsinghua Shenzhen International Graduate School, Tsinghua University, Shenzhen, China [2]Microsoft Research Asia, Microsoft, Beijing, China [3]Faculty of Computer Science and Artificial Intelligence, Shenzhen University of Advanced Technology, Shenzhen, China [4]Economics & Technology Research Institute, China National Petroleum Corporation, Beijing, China. Correspondence to: Chun Yuan <yuanc@sz.tsinghua.edu.cn>, Ye Guo <yezi414616@outlook.com>.

*Proceedings of the 43$^{rd}$ International Conference on Machine Learning*, Seoul, South Korea. PMLR 306, 2026. Copyright 2026 by the author(s).

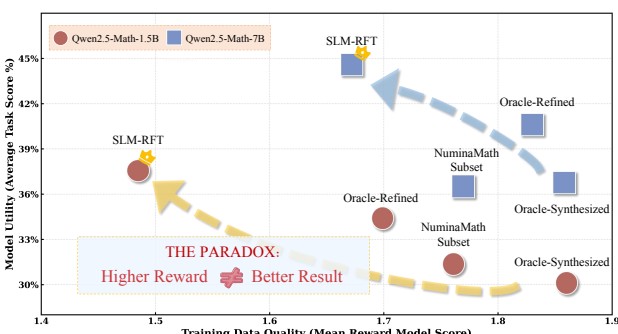

*Figure 1.* **The Quality-Utility Paradox**. We visualize the relationship between the **Perceived Quality** of training data and its **Actual Downstream Utility**. The SLM-RFT data, despite having the lowest reward scores, yields the highest performance for both 1.5B and 7B models. Conversely, data refined or generated by the Oracle achieves superior reward scores but leads to suboptimal downstream performance.

## 1. Introduction

The dominant paradigm in training Small Language Models (SLMs) for mathematical reasoning relies heavily on knowledge distillation from stronger models (Hinton et al., 2015; Gou et al., 2021). In this standard supervised fine-tuning (SFT) pipeline, a superior Synthesis Oracle (e.g., GPT-5.2) generates or refines reasoning traces, and practitioners often use reward model scores as a proxy for data quality (Lightman et al., 2023). This practice induces a data selection principle in which traces that appear more rigorous to a strong evaluator are expected to provide more useful supervision for the target SLM.

However, this assumption based on reward scores can fail for SLMs in mathematical reasoning. As illustrated in Figure 1, we identify a counterintuitive phenomenon termed the **Quality-Utility Paradox**. Training data refined by a superior Oracle obtains substantially higher reward scores, yet consistently underperforms traces generated by the SLM itself and selected through rejection sampling. The key issue is not that Oracle repair is intrinsically harmful. Rather, Oracle refinement couples two effects that current data selection criteria often conflate. It can improve the apparent logical quality of a trace while simultaneously moving that trace

away from the target model's native reasoning distribution. When the induced distributional drift raises the learner's adaptation cost, higher perceived quality need not translate into higher downstream utility.

Prior research has also begun to question whether stronger teachers or ostensibly better synthetic data always produce better students. Li et al. (2025) show that small models may fail to benefit from long reasoning traces or traces produced by strong teachers, which is consistent with our observation that improved reasoning traces can become difficult for SLMs to absorb. Bansal et al. (2025) further show that weaker generators can be preferable under matched compute budgets, highlighting that teacher strength and downstream utility need not be monotonically aligned. We therefore study a controlled mathematical reasoning setting in which the problem set is fixed and the main intervention is whether an Oracle rewrites the SLM's own reasoning trace. This design lets us isolate how logical improvement can be coupled with a shift away from the SLM's native reasoning distribution. We also note that some studies use different experimental settings yet report conclusions that appear to conflict with ours, such as parts of OpenMathInstruct-2 (Toshniwal et al., 2024). Appendix G provides a detailed reconciliation.

Our analysis therefore shifts the focus from absolute data quality to *compatibility between learner and data*. In our GPT-5.2 refinement setting, the resulting mismatch visibly manifests as **Syntactic Compaction**, where dense symbolic expressions replace the SLM's looser natural scaffolding, including explicit spacing tokens and verbal delimiters. This compaction is not claimed to be a universal failure mode of all Oracles. It is a concrete instance within mathematical reasoning where stronger models may express repaired reasoning through representations that are efficient for themselves or preferred by reward models, but costly for a smaller target model to imitate and internalize.

To test this mechanism, we introduce **Style-Aligned Refinement**, which preserves the native trajectory of the SLM while retaining logical repair from the Oracle. This intervention lowers adaptation cost and restores downstream utility, showing that Oracle improvements become useful when they are delivered through a representation compatible with the learner. Our primary contributions are as follows.

- We identify the **Quality-Utility Paradox** in SLM distillation, where data with higher reward model scores can yield inferior downstream performance compared to RFT data generated by the model itself.

- We show that Oracle refinement introduces a compatibility problem between learner and data, where logical repair can be accompanied by distributional drift that increases the target SLM's adaptation cost.

- We use **Style-Aligned Refinement** as an intervention to decouple logical rectification from stylistic drift, demonstrating that improvements from the Oracle become beneficial when they remain compatible with the target model's native reasoning distribution.

## 2. Related Work

**Knowledge Distillation in Language Models.** Knowledge distillation transfers capabilities from large teacher models to efficient student models, and has become a key method for training small language models (SLMs) in reasoning tasks (Gu et al., 2024; Xu et al., 2024b). Recent works optimize KD with objectives such as reverse KL divergence (Hamman et al., 2025) or sampling schemes based on difficulty (He et al., 2025). These methods often improve the transfer of teacher knowledge, but they do not fully characterize when high scoring teacher data is useful for a particular student. Our work studies this question from the perspective of data construction by showing that stronger Oracle traces can become less useful when logical repair is coupled with distributional drift away from the target SLM.

**On-Policy Distillation.** A growing line of on-policy distillation (OPD) work revisits the distribution mismatch in autoregressive KD by letting the student generate trajectories and receive dense teacher supervision on the states it actually visits (Agarwal et al., 2024). Subsequent studies develop practical variants of this idea, including interleaved student-teacher sampling to bridge the teacher-student gap (Xu et al., 2025), black-box OPD with adaptive feedback when teacher logits are unavailable (Ye et al., 2026), and post-training recipes that combine on-policy relevance with dense distillation signals (Lu & Lab, 2025). Mechanistic analyses further suggest that OPD success depends on compatibility between teacher and student reasoning patterns, rather than teacher strength alone (Li et al., 2026). Our work is complementary to this algorithmic line of research.

**Synthetic Data Generation and Rejection Sampling.** Synthetic data generation, often through self improvement loops in large language models (LLMs), has become a cornerstone for enhancing reasoning capabilities (Wang et al., 2023; Xu et al., 2024a). Methods such as rejection sampling fine tuning (RFT) sample multiple outputs and filter high quality ones to create training datasets (Wang et al., 2025a). Recent advancements, such as hybrid rejection sampling with preference optimization, emphasize curating diverse synthetic pairs with high reward scores to boost performance (Chen et al., 2024; Khaki et al., 2024). These studies highlight the importance of quality filtering. Our results suggest a complementary factor, since traces selected from the target SLM's own distribution can outperform higher reward Oracle variants. This indicates that rejection sampling may

benefit SLM training not only through quality filtering, but also through the learner compatible distribution induced by sampling from the student (Tian et al., 2026).

# 3. Preliminaries & Experimental Setup

We establish a controlled experimental framework to compare reasoning traces that share the same problem set but differ in how logical repair and representation form are coupled. This design lets us study how learner data compatibility affects downstream utility.

## 3.1. Model Configurations

**Target Small Language Models (SLMs).** We designate Qwen2.5-Math-1.5B (Yang et al., 2024) as the primary subject for our mechanism analysis. To evaluate whether the observed pattern extends beyond a single model, we also report results on Qwen2.5-Math-7B (Yang et al., 2024), LLaMA-3.2-3B (Grattafiori et al., 2024), and DeepSeek-Math-7B (Shao et al., 2024), as detailed in Table 15.

**Synthesis and Evaluation Oracles.** We adopt **GPT-5.2** (Bubeck et al., 2025) as the primary global Oracle for data synthesis and standard refinement tasks. We additionally employ **Qwen2.5-Math-72B-Instruct** (Yang et al., 2024) specifically for the Style-Aligned Refinement intervention.

For automated quality assessment, we employ **Qwen2.5-Math-72B-Reward** (Yang et al., 2024) as the primary evaluator. To assess robustness across evaluators, we additionally use **Skywork-Reward-Llama-3.1-27B** (Liu et al., 2024) and **Llama-3.3-Nemotron-70B-Reward** (Wang et al., 2025b) for cross model validation, with details provided in Appendix C. We refer to the scores produced by these models as **Perceived Quality**, and contrast them with **Actual Downstream Utility** measured by task accuracy.

## 3.2. Training Algorithms

To reduce the possibility that the observed performance degradation is tied to a single optimization objective, we validate our findings across two training paradigms, canonical Supervised Fine-Tuning (SFT) and robustness oriented Dynamic Fine-Tuning (DFT).

**Supervised Fine-Tuning (SFT).** We adopt the standard SFT objective, which minimizes the negative log-likelihood of the ground-truth target sequence $y^*$ conditioned on the input $x$. This serves as our baseline optimization strategy.

$$\mathcal{L}_{\text{SFT}}(\theta) = \mathbb{E}_{(x,y^*)\sim\mathcal{D}}[-\log \pi_\theta(y^*|x)] \quad (1)$$

**Dynamic Fine-Tuning (DFT).**

We employ DFT (Wu et al., 2025) to mitigate overfitting.

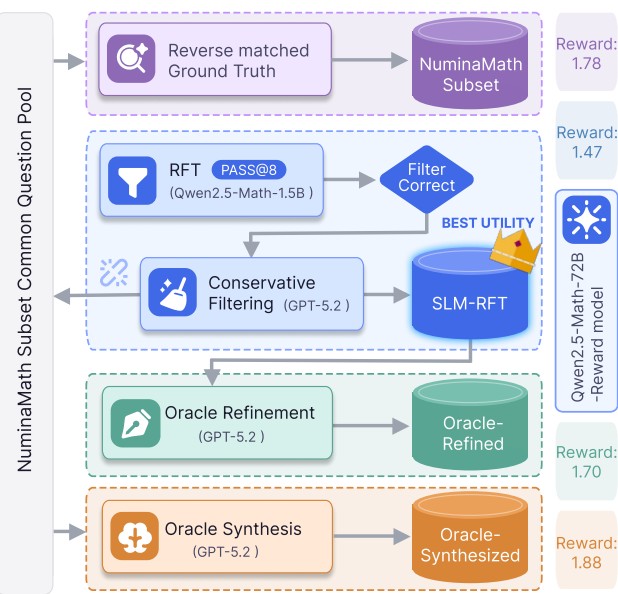

*Figure 2.* The Data Construction Pipeline. We construct four parallel datasets on an identical problem set to isolate the impact of data representation on model performance.

DFT dynamically modulates the learning signal according to model confidence by down-weighting low-confidence target tokens, thereby stabilizing training and reducing high-variance updates. The reweighted objective is formulated below.

$$\mathcal{L}_{\text{DFT}}(\theta) = \mathbb{E}_{(x,y^*)\sim\mathcal{D}}[-\text{sg}(\pi_\theta(y^*|x))\log \pi_\theta(y^*|x)] \quad (2)$$

where $\text{sg}(\cdot)$ denotes the stop-gradient operator. Reproducing the Quality-Utility Paradox under this training regime provides evidence that the observed misalignment is primarily data centric rather than an artifact of the SFT objective alone.

## 3.3. Data Construction Pipeline

**Problem Set Construction via RFT.** As illustrated in Figure 2, we initially sample 100,000 problems from the `NuminaMath CoT dataset` (Li et al., 2024), following the experimental scale established in DFT (Wu et al., 2025). We then perform Rejection Fine-Tuning (RFT) sampling using `Qwen2.5-Math-1.5B` as the SLM ($N = 8$, $T = 1.0$), where we retain one correct solution per problem without additional quality ranking among correct solutions. This process yields a subset of approximately 34,000 solvable problems. All four comparative datasets share this common problem set, ensuring that observed performance differences are not driven by problem difficulty.

Based on this fixed problem set, we construct four parallel data streams.

*Table 1.* **The Quality-Utility Misalignment.** We evaluate the impact of data representations under two fine-tuning paradigms. Despite receiving the lowest perceived quality according to the `Qwen2.5-Math-72B-Reward` model, SLM-RFT data yields stronger downstream utility than Oracle-generated alternatives with higher perceived quality under both evaluated training algorithms.

| Dataset Source | Perceived Quality (Reward) | | | Method | Downstream Utility (Avg@16 Accuracy) | | | | | Overall |
| | Mean | Std Dev | Median | | MATH500 | AIME24 | AMC23 | Minerva | Olympiad | Avg. |
|---|---|---|---|---|---|---|---|---|---|---|
| **Oracle-Synthesized** | 1.88 | 2.73 | 2.45 | SFT | 51.6 | 0.0 | 35.0 | 13.6 | 16.1 | 23.26 |
| | | | | DFT | 64.8 | 3.3 | 35.0 | 19.1 | 27.9 | 30.02 |
| **NuminaMath Subset** | 1.78 | 2.27 | 2.14 | SFT | 41.4 | 0.0 | 15.0 | 15.1 | 12.1 | 16.72 |
| | | | | DFT | 64.0 | 6.7 | 42.5 | 19.5 | 23.7 | 31.28 |
| **Oracle-Refined** | 1.70 | 2.58 | 2.22 | SFT | 46.8 | 0.0 | 22.5 | 14.0 | 14.7 | 19.60 |
| | | | | DFT | 67.8 | 10.0 | 40.0 | 22.1 | 30.4 | 34.06 |
| **SLM-RFT** | 1.47 | 3.20 | 2.13 | SFT | 49.8 | 0.0 | 35.0 | 13.2 | 15.7 | 22.74 |
| | | | | **DFT** | **70.0** | **10.0** | **40.0** | **32.0** | **33.3** | **37.06** |

- **NuminaMath Subset.** We retrieve the ground-truth CoT solutions via reverse matching from the source dataset.

- **SLM-RFT.** This dataset comprises valid solutions independently generated by the target SLM. We apply a lightweight conservative filter via the Oracle (Appendix A.1) to strip incidental generation artifacts while preserving the model's native reasoning logic and presentation style.

- **Oracle-Refined.** We construct this dataset by feeding the SLM-RFT solutions into the Oracle (`GPT-5.2`) using a targeted rectification prompt (Appendix A.2). This process is designed to repair specific imperfections, such as syntax errors and incoherent transitions, while following a principle of minimal intervention. As analyzed later, even this targeted refinement can introduce representation drift.

- **Oracle-Synthesized.** The Oracle generates these solutions directly from scratch.

### 3.4. Evaluation Protocols

We evaluate performance across a spectrum of mathematical benchmarks, spanning standard datasets (MATH500 (Hendrycks et al., 2021), Minerva Math (Lewkowycz et al., 2022)) and competition level challenges (AIME 2024, AMC 2023, OlympiadBench (He et al., 2024)). Adhering to established protocols (Wu et al., 2025), we sample 16 independent zero shot CoT generations ($T = 1.0$, max tokens 4096) and report Avg@16, the average accuracy across these generations. We also track the mean reward score of each training dataset to contrast perceived quality with observed downstream utility.

## 4. The Phenomenon of Quality-Utility Misalignment

This section establishes the Quality-Utility Paradox by comparing perceived data quality with downstream fine-tuning utility. All compared datasets are constructed on the same problem set, so the comparison controls for problem difficulty. We first report the aggregate mismatch between reward scores and downstream accuracy, and then examine whether the pattern persists across training time, model scale, and hyperparameter choices.

### 4.1. Aggregate Quality-Utility Misalignment

Table 1 shows a clear mismatch between perceived quality and downstream utility. According to the reward model, Oracle-Synthesized receives the highest mean score (1.88), followed by NuminaMath Subset (1.78), Oracle-Refined (1.70), and SLM-RFT (1.47). However, downstream accuracy does not follow this ordering. Under DFT, SLM-RFT achieves the best Avg@16 accuracy (37.06), outperforming Oracle-Refined (34.06), NuminaMath Subset (31.28), and Oracle-Synthesized (30.02). The same mismatch is also visible under standard SFT, where higher reward scores do not consistently translate into higher downstream utility.

This result defines the Quality-Utility Paradox. Data with higher perceived quality according to reward models can be less useful for fine-tuning the target SLM. Cross-model reward validation in Appendix C further shows that this reward-based ranking is not specific to a single evaluator.

### 4.2. Robustness Across Training Dynamics

We next examine whether the mismatch is a transient training artifact. The top row of Figure 3 tracks the 1.5B model on `MATH500`, `Minerva Math`, and `OlympiadBench` throughout training, together with the macro-average accuracy. SLM-RFT establishes an early lead and maintains it across the observed training trajectory, while Oracle-Refined

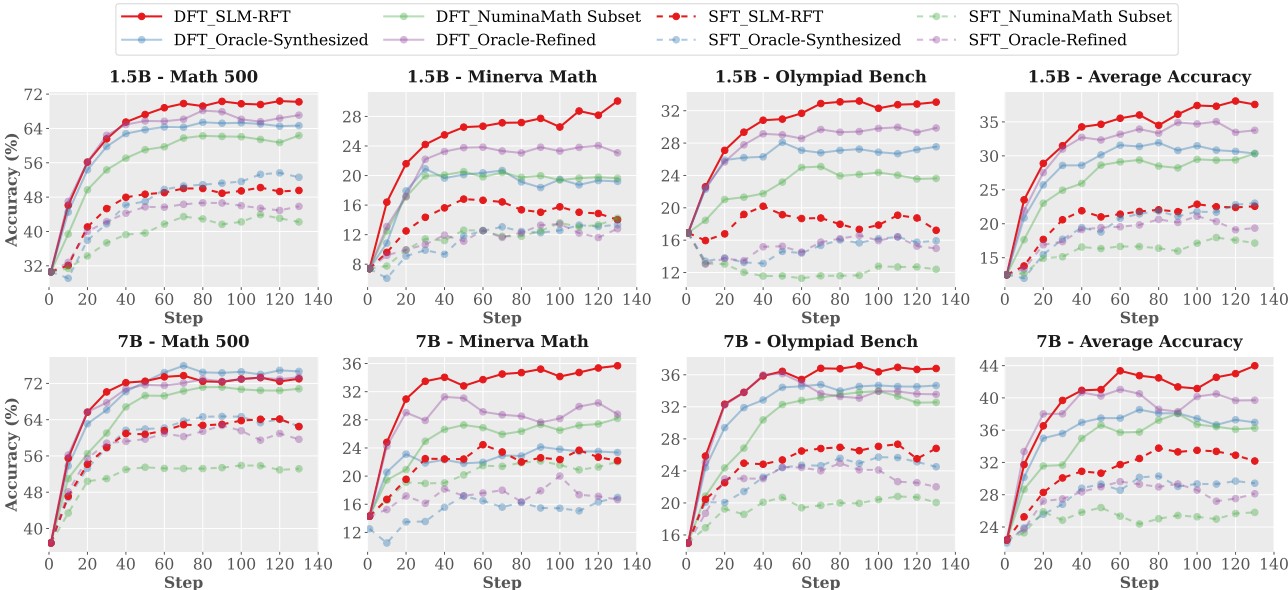

*Figure 3.* Training Dynamics across Model Scales. Validation accuracy trajectories for 1.5B (top) and 7B (bottom) models across various benchmarks. The SLM-RFT (red) remains above the other variants throughout the observed training process.

and Oracle-Synthesized remain lower. This sustained separation shows that the paradox reflects a stable pattern in learning dynamics rather than isolated advantages at a few training steps.

### 4.3. Robustness Across Model Scale

The bottom row of Figure 3 evaluates whether the same pattern holds at a larger scale. With Qwen2.5-Math-7B, SLM-RFT still outperforms Oracle-Refined, and Oracle-Synthesized remains the weakest among the compared data sources. Extended results in Appendix H further show similar behavior across LLaMA-3.2-3B and DeepSeek-Math-7B. Thus, the observed mismatch extends beyond the primary target model within our evaluated SLM setting.

### 4.4. Hyperparameter Sensitivity

Finally, we test whether the ranking depends on a particular optimization setting. Following the DFT protocol, the main experiments use learning rate 5e-5, batch size 256, and one epoch. Table 2 reports a grid search over learning rates and batch sizes on Qwen2.5-Math-1.5B. SLM-RFT remains best under all tested settings, providing evidence that the paradox is not caused by a single hyperparameter configuration.

## 5. How Oracle Refinement Alters Learning Signals

The results in Section 4 reveal a central tension. Oracle-Refined data receives higher reward scores but delivers lower downstream utility than the SLM's own generated

*Table 2.* **Hyperparameter Sensitivity on Qwen2.5-Math-1.5B.** Avg@16 accuracy on MATH under DFT with different learning rates and batch sizes.

| LR | BS | SLM-RFT | Oracle-Refined | Oracle-Synthesized |
|------|-----|---------|----------------|--------------------|
| 2e-5 | 128 | **36.16** | 34.72 | 32.96 |
| 2e-5 | 256 | **35.42** | 35.26 | 33.60 |
| 5e-5 | 128 | **39.36** | 33.96 | 30.32 |
| 5e-5 | 256 | **37.06** | 34.06 | 30.02 |

traces. To unpack this discrepancy, we examine what refinement changes beyond nominal correctness. We first analyze how Oracle refinement reshapes the token-level representation of reasoning traces, and then evaluate whether these changes preserve the SLM's original informational trajectory and how they interact with perceived quality and downstream utility.

### 5.1. Token Distribution Shift

We begin with the representation form, because the most visible effect of Oracle refinement is a shift in the local token patterns used to express mathematical reasoning.

**Token Frequency Estimation.** To quantify this shift, we tokenize each dataset with the target SLM tokenizer and estimate the unigram token distribution over all reasoning traces. For each token, we compute its normalized frequency within each dataset and compare the resulting distributions between SLM-RFT and Oracle-Refined data. Figure 4 summarizes this comparison, and full top-token statistics for all dataset variants are provided in Appendix F.

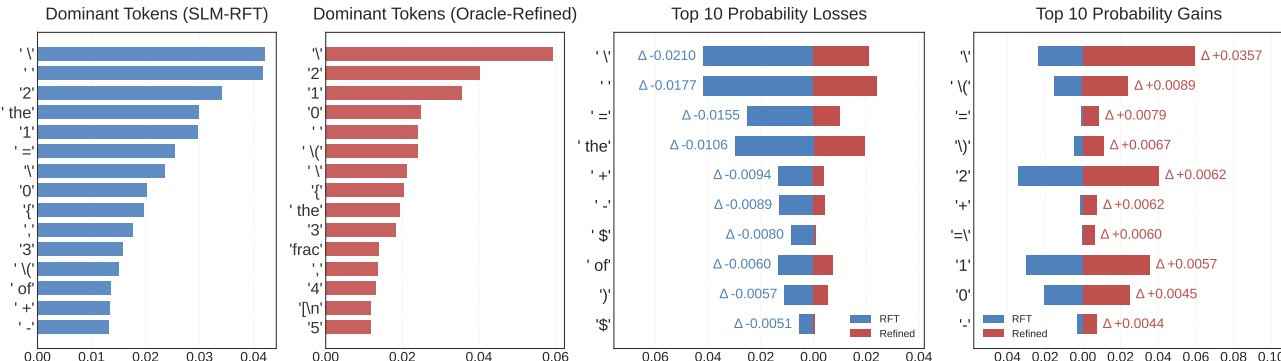

*Figure 4.* **Quantitative Distribution Shift.** The left panel shows dominant unigram token profiles of SLM-RFT and Oracle-Refined reasoning traces under the target SLM tokenizer. The right panel shows tokens with the largest probability gains and losses after refinement, revealing a systematic loss of native scaffolding tokens.

**Loss of Native Scaffolding.** The most salient negative shift is the suppression of explicit spacing and natural delimiters. Tokens that are prevalent in the SLM-RFT distribution undergo the largest reductions during refinement. For instance, the spaced backslash (' \'), which serves as a primary delimiter in SLM-RFT traces (2.1%), experiences the largest absolute decline. Similarly, natural delimiters such as ' =' and ' the' are substantially diminished. These tokens provide native scaffolding for the SLM by separating equations, connecting intermediate steps, and keeping mathematical expressions embedded in a familiar verbal structure.

**Syntactic Compaction.** The corresponding positive shift is a move toward dense, non-spaced operators. The frequency of the raw backslash ('\'), without a preceding space, increases by 3.6% and becomes the dominant token in the refined distribution. This transformation changes more than surface appearance. It collapses loosely separated reasoning steps into denser instruction chains. As illustrated in Appendix E, the resulting structural density may make the refined traces harder for the SLM to parse in the same way as its native reasoning traces.

This pattern should not be interpreted as a universal property of all Oracle models. It reflects an Oracle-specific expression bias in our refinement setting. Even under the same refinement objective and explicit style constraints, different Oracle models may retain distinct presentation habits, as shown in Appendix E. The central issue is thus not compaction itself, but the mismatch between the Oracle's preferred style and the SLM's native reasoning distribution.

### 5.2. Semantic Preservation and Downstream Utility

We use an LLM-as-a-Judge protocol to assess semantic preservation by comparing each candidate solution with the corresponding SLM-RFT solution. The resulting score measures closeness to the SLM's original informational atoms

*Table 3.* **Semantic Preservation, Perceived Quality, and Utility.** Score and Rank measure closeness to the SLM-RFT trajectory. RM reports Qwen2.5-Math-RM perceived quality, and Acc. denotes Avg@16 downstream accuracy.

| Dataset | Score | Rank | RM | Acc. |
|---|---|---|---|---|
| **Style-Aligned (Qwen)** | **4.77** | **1.44** | 1.37 | **39.12** |
| **Oracle-Refined** | 4.26 | 2.21 | 1.70 | 34.06 |
| **Style-Aligned (GPT-5.2)** | 4.07 | 2.44 | 1.81 | 38.21 |
| **Oracle-Synthesized** | 3.91 | 2.97 | **1.88** | 30.02 |

and reasoning trajectory, not absolute solution quality. Thus, a higher score indicates stronger preservation of the SLM-RFT baseline, whereas a lower score may reflect Oracle-side repair, reorganization, or polishing. For completeness, we also report our earlier embedding-based analysis in Appendix D, where we discuss why it is insufficient as the main evidence. Table 3 also includes the Style-Aligned variants introduced later in Section 6, which serve as interventions that preserve native style during logical repair.

### Judge Protocol.

We sample 3,543 paired examples and ask GPT-5.2 to compare each candidate solution against the corresponding SLM-RFT solution. The judge is instructed to ignore formatting and surface lexical variation, and to evaluate only whether the candidate preserves the original informational atoms, including variable definitions, intermediate equations, hypotheses, and trial-and-error steps. We report a five-point semantic preservation score and the average closeness rank. The full evaluation prompt is provided in Appendix A.4.

### Semantic Preservation vs. Perceived Quality.

Table 3 reveals that semantic preservation and perceived quality move in opposite directions. Style-Aligned (Qwen) is closest to the SLM-RFT trajectory, achieving the highest semantic score (4.77) and the best average rank (1.44), but receives the lowest reward-model score (1.37). In contrast,

Oracle-Synthesized is farthest from SLM-RFT (Score 3.91, Rank 2.97) while receiving the highest reward score (1.88). This contrast is consistent with the role of Oracle-side repair, reorganization, and polishing, which can improve perceived logical quality while moving the data away from the SLM-native trajectory.

**Implication for Downstream Utility.**

The reward scores confirm the intuitive benefit of Oracle intervention. Lower semantic preservation often reflects stronger repair, reorganization, or polishing of the original SLM trajectory, which improves perceived data quality. However, downstream utility follows a different principle. Data that remains closer to the SLM-native trajectory provides a pseudo-on-policy learning signal that the target model can internalize more easily. This compatibility can dominate perceived quality itself. Style-Aligned (Qwen) receives the lowest reward score (1.37) but achieves the highest Avg@16 accuracy (39.12), whereas Oracle-Refined obtains a substantially higher reward score (1.70) but reaches only 34.06 accuracy. The Style-Aligned (GPT-5.2) variant further supports this trend. Although it is less faithful than the Qwen variant, its partial restoration of the native trajectory still improves over standard Oracle-Refined data (38.21 vs. 34.06).

This analysis clarifies the origin of the Quality-Utility Paradox. Oracle refinement introduces two opposing forces. On the one hand, it repairs, reorganizes, and polishes the original reasoning trajectory, thereby improving perceived data quality in a way that should, under the standard intuition, benefit downstream learning. On the other hand, the same refinement process also injects the Oracle model's own stylistic preferences and shifts the reasoning traces away from the SLM-native distribution. Our results show that, at least in the SLM regime, this pseudo-on-policy distributional compatibility is more important than perceived data quality itself.

## 6. The Adaptation Cost of Oracle Refinement

Section 5 shows that Oracle refinement introduces two opposing forces. It repairs and polishes reasoning traces, thereby improving perceived quality, but it also shifts those traces away from the SLM-native distribution. We now ask why the latter effect can dominate the former during fine-tuning. To this end, we move from data-level statistics to target-model learnability, using perplexity to quantify the adaptation cost imposed by different data representations. We then test the resulting mechanism through a targeted Style-Aligned intervention that preserves the native trajectory while retaining Oracle-side logical repair.

*Table 4.* Perplexity Distribution Analysis. Global PPL follows a monotonic negative association with task performance. This hierarchy remains stable across all sequential segments ($Q_1 \sim Q_4$).

| Dataset | Global PPL ($\downarrow$) | Segmented PPL($\downarrow$) | | | | Task Score ($\uparrow$) |
|---|---|---|---|---|---|---|
| | | $Q_1$ | $Q_2$ | $Q_3$ | $Q_4$ | |
| **SLM-RFT** | **1.52** | **1.53** | 1.31 | 1.28 | 1.32 | **37.06** |
| **Oracle-Refined** | 1.85 | 1.92 | 1.73 | 1.68 | 1.80 | 34.06 |
| **NuminaMath Subset** | 1.89 | 1.97 | 1.59 | 1.54 | 1.67 | 31.28 |
| **Oracle-Synthesized** | 2.69 | 3.28 | 2.36 | 2.22 | 2.16 | 30.02 |

### 6.1. Perplexity and Downstream Utility

To quantify the target SLM's adaptation overhead, we use Global Perplexity (PPL) as a proxy for distributional compatibility. Here PPL is not intended to measure absolute data quality. Instead, it measures how predictable a training trace is under the target SLM, and therefore reflects the adaptation burden associated with its representation form. We complement Global PPL with Segmented Perplexity ($Q_1$ to $Q_4$) for fine-grained validation. Specifically, we partition each reasoning trajectory into four quartiles of equal length, where $Q_1$ approximates the initiation phase and $Q_4$ captures the concluding steps.

**Validation of Global PPL.** As shown in Table 4, Global PPL follows a monotonic negative association with downstream performance. The SLM-RFT dataset, which is closest to the model's native distribution, achieves the lowest Global PPL (1.52) and the highest task accuracy (37.06%). In contrast, Oracle-Refined (1.85), NuminaMath Subset (1.89), and Oracle-Synthesized (2.69) impose progressively higher adaptation costs and yield lower downstream utility. This hierarchy remains stable across all quartiles ($Q_1 \sim Q_4$), indicating that the advantage of native-distribution data is not a local artifact of a particular reasoning stage. Rather, the target SLM finds pseudo-on-policy traces easier to process throughout the full reasoning trajectory.

**The Adaptability Gap in $Q_1$.** Although the relative difficulty ranking remains consistent across all segments, the composition of the processing burden is most visible during the initiation phase ($Q_1$). As shown in Table 4, all datasets exhibit their highest perplexity near the beginning of the sequence, where the model must establish an initial reasoning trajectory. However, the token-level attribution in Table 5 reveals a qualitative difference in what the model is asked to predict. For SLM-RFT, the primary loss contributors are natural language initiators such as 'To' and 'the', which are tied to ordinary reasoning setup. For Oracle-Refined data, the $Q_1$ region is dominated by structural syntactic markers such as '\('. The refined traces therefore introduce Oracle-specific presentation constraints before the mathematical reasoning itself can be internalized.

*Table 5.* Detailed Q1 Loss Attribution. We contrast the top 12 loss contributors in the start-of-sequence phase. SLM-RFT is driven by semantic initiators, while Oracle-Refined is dominated by syntactic markers.

| Rank | SLM-RFT | | | Oracle-Refined | | |
|---|---|---|---|---|---|---|
| | Token | Freq | NLL | Token | Freq | NLL |
| 1 | 'To' | 0.6% | 2.53 | '\(' | 2.3% | 0.99 |
| 2 | 'the' | 4.4% | 0.29 | '\' | 4.6% | 0.43 |
| 3 | 'Let' | 0.2% | 5.05 | 'To' | 0.7% | 2.50 |
| 4 | '$' | 1.0% | 1.08 | 'the' | 2.7% | 0.57 |
| 5 | 'The' | 0.4% | 1.76 | '\' | 2.1% | 0.65 |
| 6 | ',' | 2.2% | 0.28 | 'Let' | 0.2% | 5.10 |
| 7 | '\(' | 1.9% | 0.31 | '=\' | 0.7% | 1.50 |
| 8 | '\' | 1.5% | 0.37 | 'use' | 0.1% | 8.11 |
| 9 | '\n' | 0.2% | 3.11 | '\n\n' | 0.1% | 7.57 |
| 10 | '\' | 3.6% | 0.14 | '(' | 0.4% | 2.20 |
| 11 | '.' | 1.4% | 0.30 | '1' | 3.3% | 0.20 |
| 12 | ' ' | 2.9% | 0.14 | ',' | 1.6% | 0.37 |

This provides model-side evidence for the distributional mismatch identified in Section 5.1.

### 6.2. Mechanism Validation through Style Alignment

The preceding analyses suggest a causal hypothesis. The weakness of Oracle-Refined data is not caused by the absence of useful logical repair, but by the adaptation cost introduced when this repair is expressed in an Oracle-specific distribution. If this hypothesis is correct, then decoupling logical correction from stylistic drift should reduce adaptation cost and restore downstream utility.

**Experimental Intervention.**

We introduce a targeted intervention termed **Style-Aligned Refinement** (prompts detailed in Appendix A.3). In contrast to standard refinement that enforces rigid formal presentation norms, this strategy instructs the Oracle to rectify logical errors while strictly emulating the SLM's native linguistic style. This approach prioritizes *distributional compatibility*, keeping the reasoning traces close to the SLM's learnable distribution.

**Lower Cost and Lower Reward.** The global perplexity analysis in Table 6 shows a clear reduction in adaptation cost for Style-Aligned data compared to the standard Oracle-Refined baseline. Notably, the Qwen-based refinement variant achieves a PPL of **1.46**, falling below the native SLM-RFT baseline of 1.52. As qualitatively analyzed in Appendix E, this improvement stems from the refiner's ability to preserve the SLM's structural scaffolding while correcting reasoning inconsistencies. By filtering out incidental generation noise without imposing an unfamiliar presentation style, the intervention yields traces that are more predictable for the target model than its own raw generations.

*Table 6.* **Mechanism Validation.** Comparison of adaptation overhead (PPL) and perceived quality (Reward). The Style-Aligned (Qwen) strategy achieves the lowest perplexity but receives the lowest reward score, highlighting the mismatch between reward-based quality estimates and native-distribution learnability.

| Dataset Variant | Global PPL ($\downarrow$) | Reward Score ($\uparrow$) |
|---|---|---|
| NuminaMath Subset | 1.89 | 1.78 |
| Oracle-Synthesized | 2.69 | 1.88 |
| Oracle-Refined | 1.85 | 1.70 |
| SLM-RFT | 1.52 | 1.47 |
| Style-Aligned (GPT-5.2) | 1.78 | 1.81 |
| **Style-Aligned (Qwen)** | **1.46** | 1.37 |

Evaluations with the Qwen2.5-Math-RM further expose the divergence between perceived quality and learnability. While Oracle-Refined data retains a high reward score of 1.70, the Style-Aligned (Qwen) variant receives the lowest score of 1.37 among all candidates, even below the SLM-RFT baseline of 1.47. This result does not imply that Style-Aligned data is logically worse. Rather, it shows that reward models can undervalue native-distribution traces whose presentation remains informal but highly learnable for the target SLM.

**Outcome.**

The fine-tuning results in Figure 5 provide mechanism-level evidence. By mitigating distributional misalignment, the **Style-Aligned (Qwen)** model achieves a peak accuracy of **39.12**, surpassing both the standard Oracle-Refined counterpart (34.06) and the native SLM-RFT baseline (37.06). This indicates that Oracle-side logical repair is beneficial when it is delivered through a representation that the target SLM can readily internalize.

The GPT-5.2 Style-Aligned variant provides a complementary check. Although its distribution remains less native than the Qwen-aligned variant, it still maintains a clear advantage over standard Oracle-Refined data for most of the training process. This partial recovery supports the same mechanism. Reducing stylistic drift improves the SLM's ability to convert Oracle refinement into downstream gains, even when the alignment is imperfect.

## 7. Conclusion

We identify the *Quality-Utility Paradox*, where training data with higher perceived quality can yield lower downstream utility for SLM mathematical reasoning. Our analysis shows that Oracle refinement introduces two opposing effects. It repairs and polishes reasoning traces, improving reward-model scores, but it also shifts the data away from the SLM's native reasoning distribution through Oracle-specific stylistic preferences. This distributional drift increases the target model's adaptation cost and can outweigh the benefit of

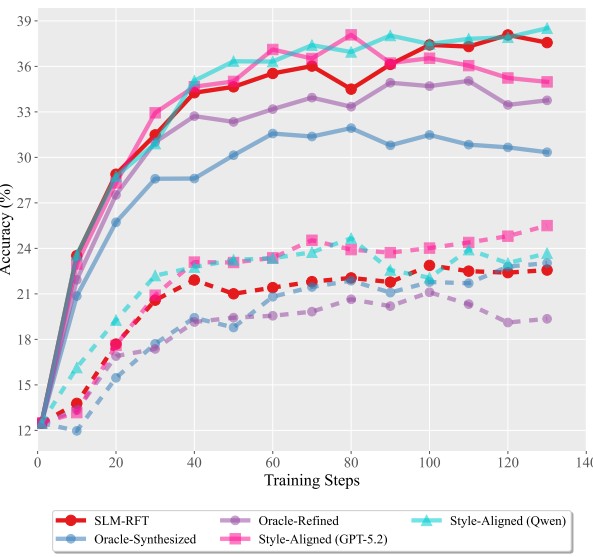

*Figure 5.* Dynamics of Style-Aligned Training. Style-Aligned Qwen surpasses all baselines, showing that reducing syntactic mismatch allows the model to benefit more effectively from logical refinement. Style-Aligned GPT also improves over standard Oracle-Refined data for most of training, further supporting the importance of distributional compatibility.

improved logical presentation. Style-Aligned Refinement preserves the SLM-native trajectory while retaining logical repair, showing that Oracle improvements are most useful when they remain learnable for the target model. These findings suggest that effective distillation for SLMs should optimize not only perceived data quality, but also distributional compatibility with the learner.

## 8. Limitations

Our study focuses on mathematical reasoning with SLMs. Although we evaluate multiple model families and scales, the evidence remains confined to the SLM regime and math-centric training data. Whether the same quality-utility trade-off holds for larger models, non-mathematical domains, or mixed-task instruction tuning remains an open question. In addition, Style-Aligned Refinement is used primarily as a prompt-based intervention to test our mechanism. Practical deployment may require more systematic approaches, such as automated style transfer, learner-aware data filtering, or reward models that explicitly account for the target model's distributional compatibility.

## Acknowledgments

This work was partially conducted during Hao-long Qian's internship at Microsoft Research Asia. This work was supported by the SSTIC Grants KJZD20230923115106012, KJZD20230923114916032, and GJHZ20240218113604008.

## Impact Statement

This work studies how synthetic reasoning data affects SLM training in mathematical tasks. Its primary impact is methodological. Reward-model scores alone may be insufficient for selecting useful distillation data, and learner-aware distributional compatibility should be considered alongside logical quality. We do not anticipate direct negative societal impacts beyond those generally associated with improving the reasoning ability of language models.

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

# A. Prompt Templates

This appendix provides the exact system prompts used in our data construction pipeline. Unless otherwise specified, prompts were executed using **GPT-5.2** as the Synthesis Oracle.

## A.1. Conservative Filter Prompt

The following prompt supports the **SLM-RFT** construction phase. Its primary objective is to remove completion artifacts (e.g., redundant headers or introductory filler) while preserving the SLM's original reasoning logic and distributional characteristics.

---

**System Prompt for Conservative Filtering**

```
You are a Data Quality & Formatting Specialist for Math CoT datasets.
I will provide you with a raw (Question, Solution, Ground Truth) triplet.
Your task is to CLEAN completion artifacts and FORMAT the final answer, strictly
adhering to the original logic.

**CORE PRINCIPLE: CONSERVATIVE REPAIR**
- **DO NOT** fix the model's reasoning logic. If the reasoning is very bad,
  output "DISCARD".
- **DO NOT** force-fit the answer to the Ground Truth if the calculated value
  is different.
- **ONLY** modify the final step to map a *correct* value to its corresponding
   Option.

---

### RULE 1: REMOVE COMPLETION ARTIFACTS (Strict Start-Cleaning)
The input solution is generated by a model completing the prompt "Question: ...
Answer:". This often creates "bridging" artifacts at the very beginning. You must
strip these to reveal the clean reasoning path.

**Target Patterns to Delete (at the start only):**
1.  **Premature Answers/Numbers**: Isolated numbers or letters followed by a
    newline.
    * *Example:* "3 / 10\nTo solve..." -> Keep "To solve..."
    * *Example:* "C\n\nFirst, let's..." -> Keep "First, let's..."
    * *Example:* Input `\[ MN = R \]\nLet's analyze...` -> Keep `Let's analyze...`
2.  **Placeholder Symbols**: Underscores, empty parentheses, or dotted lines.
    * *Example:* "______ To determine..." -> Keep "To determine..."
    * *Example:* "( )\n\nWe need to..." -> Keep "We need to..."
3.  **Redundant Headers**: "Answer:", "Solution:", "Step-by-step:", "Here is the
    logic:".

---

### RULE 2: BOXED FINAL ANSWER
The final line MUST be the boxed answer value.
- **Scenario 1 (Match)**: Model derives a value that matches the GT.
   -> Ensure `\boxed{Value}`.
- **Scenario 2 (Mismatch)**: Model derives a wrong value. -> **DISCARD**.
- **Scenario 3 (Format Hallucination)**: Model outputs a letter like `\boxed{C}`
  for a free response question. -> **DISCARD** (Do not guess the value).
- **Scenario 4  Equation/Proof Type**: GT is "MN = R" or "x = 3y".
  -> Box must be `\boxed{MN = R}`. (Do not just box `R` if GT is the full relation).

---

### RULE 3: STANDARD LATEX
- Ensure math is wrapped in `\$` or `\$\$`.
- Ensure the final boxed answer is the very last part of the string.
```

```
---

### RULE 4: REASONING DEPTH CHECK (The "No-CoT" Filter)
Chain-of-Thought data requires *steps*. If the solution is a "Direct Answer",
output **DISCARD**.

- **DISCARD IF**: The solution is 1-2 sentences long and contains NO derivation
    steps.
    * *Bad Example:* "The sum of the infinite series is \boxed{1/4}."
      (No math steps -> DISCARD)
    * *Bad Example:* "\boxed{-1 < a < 1}" (No context -> DISCARD)
    * *Bad Example:* "The answer is clearly B because it matches the formula."
      (Trivial/Lazy -> DISCARD)

---

### EXECUTION LOGIC:
Input Question: {question}
Input Ground Truth: {ground_truth}
Input Solution: {solution}

Output format:
- If the solution is logically consistent and matches the GT (after allowed
  formatting fixes): Output ONLY the clean solution string.
- If DISCARD condition met: Output string "DISCARD".
```

### A.2. Surgical Refinement Prompt

The following prompt is used to generate the **Oracle-Refined** dataset. It instructs the Oracle to perform targeted repairs on logic and formatting, producing data with higher perceived quality under reward-model evaluation.

```
System Prompt for Surgical Refinement

You are a Data Refinement Specialist for mathematical reasoning.
Your task is to perform a **SURGICAL REPAIR** on a solution generated by a weaker
    model.

### OBJECTIVE:
Your goal is to refine a solution generated by a weaker model (Base Model) to
enhance the reasoning path's logical flow and coherence while ensuring correctness
and strictly PRESERVING its original distribution characteristics. Even if the
final answer is correct, identify and address suboptimal elements in the reasoning,
such as unclear steps or minor gaps, with targeted improvements.

### INPUT DATA:
1. **Problem**: The math problem definition.
2. **Draft Solution**: The solution generated by the Base Model, which may contain
    formatting errors (garbled text), logical gaps, unclear phrasing, or minor
    calculation mistakes.

### YOUR TASK:
Review the [Draft Solution] and output a [Refined Solution] adhering to the
following rules:
1. **Correction**: Fix any LaTeX syntax errors, garbled characters, or Python code
    errors.
2. **Logic and Flow Optimization**: Ensure the reasoning path is coherent, logically
    sound. If there are gaps, ambiguities, or inefficient steps in the reasoning,
    make minimal adjustments to clarify or bridge them--such as rephrasing for
    better flow or adding brief connective explanations.
3. **Minimal Modification (CRITICAL)**:
    - Do NOT rewrite the entire solution; focus only on targeted fixes to improve
      readability and logical progression while keeping changes as small as possible.
    - Refining only to remove noise, errors, or minor inefficiencies in the
      reasoning path.
4. **Format**: The output must be the raw text of the solution (including logical
    steps and code blocks if present), ready to be inserted back into a JSON field.

### OUTPUT FORMAT:
Output ONLY the refined solution text. Do not include introductory phrases like
"Here is the fixed solution" or markdown code fences (```) around the entire output
(unless the solution itself contains code blocks).
-----------------------------------------------
Problem: {problem}
Draft Solution:
{solution}
-----------------------------------------------
```

### A.3. Style-Aligned Prompt

This prompt defines our Style-Aligned intervention. It instructs the Oracle to rectify logical errors while preserving the student model's native syntax and spacing to reduce *syntactic adaptation cost*.

```
You are a "Peer Correction System" for a specific Small Language Model.
Your goal is to perform an **Invisible Correction** on the student's solution while
    mimicking the student's native linguistic style.

### THE LOGIC: WHY THIS MATTERS
The student model learns poorly from "textbook quality" data because it has a
    "Syntactic Bottleneck."
- **Standard LaTeX (Compact)** = High Cognitive Load (Bad).
- **Spaced-out, Natural Math (Loose)** = Low Cognitive Load (Good).

### YOUR TASK:
- Content: Fix math/logic errors in the [Draft Solution].
- Style: Keep the language style indistinguishable from the original draft.
- Strict Formatting: You MUST ensure the final answer is placed within \boxed{} at
    the very end of the response.

### INPUT DATA:
Problem: ${problem}
Draft Solution:
${solution}

### OUTPUT:
Output ONLY the raw text of the refined solution.
```

## A.4. LLM-as-a-Judge Semantic Evaluation Prompt

This prompt is used for the semantic preservation evaluation in Section 5.2. The judge compares each candidate against the SLM-RFT answer and is explicitly instructed to ignore formatting, lexical variation, and final-answer correctness.

```
<|im_start|>system
You are an expert evaluator of **Semantic Content** in mathematical reasoning. Your
    task is to measure how faithfully each candidate preserves the exact
    informational content of the Standard Answer. Ignore correctness, final answer,
    formatting, LaTeX style, spacing, synonyms, verbosity, line breaks, and
    connective phrases.

**Semantic Content** = concrete informational atoms only: specific variable
    definitions, exact intermediate equations/derivations, qualitative
    observations, particular trial hypotheses, unique conceptual framings, and any
    distinctive explanatory choices present in the Standard Answer.

**Strict Rules**:
- Any rewording that conveys the exact same information = perfect match.
- Minor reordering of non-dependent atoms = zero penalty.
- Adding any new atom ("Key elements", "Self-check", "Self-verify", extra
    verification step) = divergence.
- Omitting any atom present in Standard = divergence.

**Scoring Rubric (1.0-5.0 continuous)**:
- 5.0: All atoms preserved exactly; differences are only rephrasing or cosmetic.
- 4.0-4.9: Highly faithful. Most atoms preserved; minor additions or omissions do
    not significantly affect the core semantics.
- 3.0-3.9: Multiple noticeable additions/omissions.
- 2.0-2.9: Significant new atoms or key omissions.
- 1.0-1.9: Mostly new atoms.
```

```
**Output Process (strict order)**:
1. "semantic_checkpoints": Extract 6-12 specific informational atoms from the
   Standard Answer ONLY. Format: "Atom X: [precise description]".
2. "checkpoint_mapping": For each candidate, describe preserved/omitted/added atoms
   and end with count (e.g., "10/11 preserved, 1 added self-check"). Keep each
   description under 25 words.
3. "scores": Final decimal scores.

Output **only** valid JSON, nothing else:
{
    "semantic_checkpoints": "Atom 1: ..., Atom 2: ...",
    "checkpoint_mapping": {
        "candidate_1": "...",
        "candidate_2": "...",
        "candidate_3": "...",
        "candidate_4": "...",
        "candidate_5": "...",
    },
    "scores": {
        "candidate_1": float,
        "candidate_2": float,
        "candidate_3": float,
        "candidate_4": float,
        "candidate_5": float,
    }
}
<|im_end|>
<|im_start|>user
----------------------------
Question:
{question}
----------------------------
Standard Answer:
{standard_answer}
----------------------------
Candidate Answer 1:
{candidate_1}
----------------------------
Candidate Answer 2:
{candidate_2}
----------------------------
Candidate Answer 3:
{candidate_3}
----------------------------
Candidate Answer 4:
{candidate_4}
----------------------------
Candidate Answer 5:
{candidate_5}
<|im_end|>
<|im_start|>assistant
```

## B. Detailed Q1 Attribution Analysis

In Section 6.1, we identified the $Q_1$ region as the point where the adaptation gap is most visible. Here, we provide a token-level breakdown for the top 15 loss contributors in this region across all four datasets.

The columns in the following tables have the following meanings.

- **Rank** gives the order by total loss contribution.

- **ID** gives the token ID in the tokenizer vocabulary.

- **Token** gives the decoded string, with ′...′ used to reveal whitespace.

- **Count** gives the number of times this token appeared in the $Q_1$ region during evaluation.

- **Freq (%)** gives the relative frequency of this token within the $Q_1$ region.

- **Avg Loss** gives the average negative log-likelihood (NLL).

- **PPL** gives the local perplexity $(\exp(\text{Avg Loss}))$ for this token.

- **Total Loss** gives the cumulative loss contribution.

We present the detailed statistics for each dataset in the tables below.

*Table 7.* **NuminaMath Subset ($Q_1$ Region).** Full statistics for the top loss contributors.

| Rank | ID | Token | Count | Freq (%) | Avg Loss | PPL | Total Loss |
|---|---|---|---|---|---|---|---|
| 1 | 400 | ′\$′ | 17514 | 2.19% | 1.1050 | 3.02 | 19352.96 |
| 2 | 279 | ′the′ | 30644 | 3.84% | 0.4193 | 1.52 | 12850.44 |
| 3 | 11 | ′,′ | 18676 | 2.34% | 0.4962 | 1.64 | 9266.46 |
| 4 | 10061 | ′Let′ | 1741 | 0.22% | 4.0682 | 58.45 | 7082.80 |
| 5 | 16 | ′1′ | 21554 | 2.70% | 0.2982 | 1.35 | 6427.41 |
| 6 | 2303 | ′\n′ | 1000 | 0.13% | 6.1125 | 451.45 | 6112.48 |
| 7 | 57960 | ′\$\′ | 3681 | 0.46% | 1.5409 | 4.67 | 5672.00 |
| 8 | 1124 | ′\′ | 27329 | 3.42% | 0.2061 | 1.23 | 5632.65 |
| 9 | 510 | ′:\n′ | 4373 | 0.55% | 1.2526 | 3.50 | 5477.54 |
| 10 | 220 | ′ ′ | 26302 | 3.29% | 0.2013 | 1.22 | 5294.60 |
| 11 | 1249 | ′To′ | 2348 | 0.29% | 2.0981 | 8.15 | 4926.22 |
| 12 | 323 | ′and′ | 6998 | 0.88% | 0.6975 | 2.01 | 4881.19 |
| 13 | 334 | ′**′ | 1594 | 0.20% | 2.9898 | 19.88 | 4765.81 |
| 14 | 12549 | ′Since′ | 1051 | 0.13% | 4.3840 | 80.16 | 4607.61 |
| 15 | 382 | ′.\n\n′ | 3318 | 0.42% | 1.1730 | 3.23 | 3892.08 |

*Table 8.* **SLM-RFT Dataset** ($Q_1$ **Region**). Full statistics for the top loss contributors. The leading tokens correspond mainly to natural language reasoning initiators.

| Rank | ID | Token | Count | Freq (%) | Avg Loss | PPL | Total Loss |
|---:|---:|---|---:|---:|---:|---:|---:|
| 1 | 1249 | 'To' | 6185 | 0.62% | 2.5286 | 12.54 | 15639.30 |
| 2 | 279 | 'the' | 44044 | 4.38% | 0.2943 | 1.34 | 12961.15 |
| 3 | 10061 | 'Let' | 2382 | 0.24% | 5.0518 | 156.30 | 12033.37 |
| 4 | 400 | '\$' | 10175 | 1.01% | 1.0800 | 2.94 | 10989.36 |
| 5 | 785 | 'The' | 4142 | 0.41% | 1.7580 | 5.80 | 7281.57 |
| 6 | 11 | ',' | 22376 | 2.23% | 0.2806 | 1.32 | 6278.02 |
| 7 | 17767 | '\(' | 18829 | 1.87% | 0.3058 | 1.36 | 5757.04 |
| 8 | 59 | '\' | 15504 | 1.54% | 0.3681 | 1.44 | 5706.61 |
| 9 | 2303 | '\n' | 1664 | 0.17% | 3.1067 | 22.35 | 5169.62 |
| 10 | 1124 | '\' | 35969 | 3.58% | 0.1398 | 1.15 | 5028.28 |
| 11 | 13 | '.' | 14452 | 1.44% | 0.3001 | 1.35 | 4336.62 |
| 12 | 220 | ' ' | 29180 | 2.90% | 0.1414 | 1.18 | 4125.96 |
| 13 | 19324 | '`\n\n' | 415 | 0.04% | 9.8726 | 19391.89 | 4097.13 |
| 14 | 323 | 'and' | 8679 | 0.86% | 0.4706 | 1.60 | 4083.97 |
| 15 | 374 | 'is' | 9296 | 0.92% | 0.4069 | 1.50 | 3782.66 |

*Table 9.* **Oracle-Refined Dataset** ($Q_1$ **Region**). Full statistics for the top loss contributors. Syntactic markers such as \ ( and \ account for a larger share of the loss.

| Rank | ID | Token | Count | Freq (%) | Avg Loss | PPL | Total Loss |
|---:|---:|---|---:|---:|---:|---:|---:|
| 1 | 17767 | '\(' | 18832 | 2.33% | 0.9907 | 2.69 | 18656.17 |
| 2 | 59 | '\' | 37267 | 4.61% | 0.4293 | 1.54 | 15998.27 |
| 3 | 1249 | 'To' | 5535 | 0.69% | 2.5006 | 12.19 | 13840.92 |
| 4 | 279 | 'the' | 21547 | 2.67% | 0.5720 | 1.77 | 12324.93 |
| 5 | 1124 | '\' | 16936 | 2.10% | 0.6543 | 1.92 | 11080.52 |
| 6 | 10061 | 'Let' | 1768 | 0.22% | 5.0982 | 163.73 | 9013.63 |
| 7 | 34433 | '=\' | 5652 | 0.70% | 1.5013 | 4.49 | 8485.44 |
| 8 | 990 | 'use' | 978 | 0.12% | 8.1136 | 3339.50 | 7935.08 |
| 9 | 19324 | '`\n\n' | 959 | 0.12% | 7.5746 | 1948.03 | 7264.02 |
| 10 | 320 | '(' | 3053 | 0.38% | 2.2048 | 9.07 | 6731.24 |
| 11 | 16 | '1' | 26692 | 3.31% | 0.2044 | 1.23 | 5457.11 |
| 12 | 11 | ',' | 12690 | 1.57% | 0.3660 | 1.44 | 4644.51 |
| 13 | 220 | ' ' | 16562 | 2.05% | 0.2762 | 1.32 | 4574.77 |
| 14 | 510 | ':\n' | 2118 | 0.26% | 2.0778 | 7.99 | 4400.76 |
| 15 | 2303 | '\n' | 1002 | 0.12% | 4.3654 | 78.68 | 4374.14 |

*Table 10.* **Style-Aligned GPT-5.2 Dataset** ($Q_1$ **Region).** Full statistics for the top loss contributors. Some tokens associated with auxiliary explanation templates receive large loss values.

| Rank | ID | Token | Count | Freq (%) | Avg Loss | PPL | Total Loss |
|---|---|---|---|---|---|---|---|
| 1 | 5309 | 'Key' | 6889 | 0.69% | 12.6406 | 308848.88 | 87081.14 |
| 2 | 5424 | 'elements' | 8218 | 0.82% | 5.5861 | 266.69 | 45906.55 |
| 3 | 17767 | '\(' | 33002 | 3.29% | 1.2792 | 3.59 | 42217.62 |
| 4 | 25 | ':' | 16699 | 1.67% | 2.3573 | 10.56 | 39364.03 |
| 5 | 20869 | 'MC' | 2756 | 0.28% | 9.4692 | 12954.63 | 26097.14 |
| 6 | 10115 | 'Self' | 2768 | 0.28% | 9.3650 | 11672.88 | 25922.39 |
| 7 | 59 | '\' | 57461 | 5.74% | 0.4326 | 1.54 | 24856.40 |
| 8 | 16 | '1' | 39103 | 3.90% | 0.5406 | 1.72 | 21138.35 |
| 9 | 1124 | '\' | 18276 | 1.82% | 1.0295 | 2.80 | 18815.46 |
| 10 | 320 | '(' | 5697 | 0.57% | 3.2009 | 24.56 | 18235.66 |
| 11 | 458 | 'an' | 3231 | 0.32% | 5.1268 | 168.48 | 16564.71 |
| 12 | 1592 | 'Key' | 1353 | 0.14% | 11.7347 | 124825.19 | 15877.01 |
| 13 | 568 | ').' | 6658 | 0.66% | 2.3546 | 10.53 | 15676.65 |
| 14 | 21693 | 'Unknown' | 1580 | 0.16% | 9.8878 | 19689.11 | 15622.76 |
| 15 | 1096 | 'This' | 1846 | 0.18% | 8.2855 | 3965.95 | 15295.03 |

# C. Robustness Analysis of Perceived Quality

To examine whether the *Quality-Utility Paradox* depends on a specific scoring metric, we conduct a cross-model evaluation using the reward models described in Section 3. This analysis tests whether the perceived-quality ranking remains stable across different evaluator architectures and scales.

## C.1. Cross-Model Evaluation Results

Table 11 reports the comparative assessments across independent evaluators. These results show that the disconnect between perceived quality and downstream utility is not specific to the primary reward model.

**Lower Ratings for Native Distributions.** A consistent ranking pattern emerges across all three reward models. The **SLM-RFT** dataset is assigned to the lower tier (Rank 5–6), despite achieving the highest utility among the original data sources (Avg@16 37.06%). Similarly, **Style-Aligned (Qwen)**, which achieves the highest overall downstream utility (39.12%), receives lower perceived-quality scores (Rank 4–6). This pattern suggests that current reward models can undervalue the verbose, granular syntactic patterns that appear useful for SLM learning.

Collectively, these findings provide additional evidence for the Quality-Utility Paradox. The high-reward signal appears more aligned with the Oracle's preferred output distribution than with the student model's downstream learning utility.

*Table 11.* **Cross-Model Quality Consistency.** Comparison of mean reward scores assigned by the independent evaluators defined in the main text. While the absolute scores vary, the variants with the highest downstream utility (**SLM-RFT** and **Style-Aligned (Qwen)**) are rated in the lower tiers across all reward models, providing additional evidence for the robustness of the paradox.

| Dataset | Avg. Tokens | Qwen2.5-Math-RM | | Skywork-RM-27B | | Nemotron-70B-RM | |
|---|---|---|---|---|---|---|---|
| | | Score | Rank | Score | Rank | Score | Rank |
| Oracle-Synthesized | 416.83 | 1.88 | 1 | -8.78 | 3 | 0.8525 | 5 |
| NuminaMath Subset | 326.73 | 1.78 | 3 | -8.81 | 4 | 1.4037 | 3 |
| Oracle-Refined | 334.91 | 1.70 | 4 | **-6.85** | **1** | 1.6410 | 2 |
| Style-Aligned (GPT-5.2) | 348.33 | 1.81 | 2 | -7.20 | 2 | **1.7391** | **1** |
| Style-Aligned (Qwen) | 412.37 | 1.37 | 6 | -9.31 | 5 | 1.2058 | 4 |
| SLM-RFT | 413.00 | 1.47 | 5 | -9.64 | 6 | 0.8499 | 6 |

## D. Auxiliary Embedding-Based Semantic Visualization

In Section 5.2, we use an LLM-as-a-Judge evaluation as the primary evidence for semantic preservation. Here, we report the earlier embedding-based analysis as an auxiliary diagnostic. These metrics provide a coarse view of distributional proximity, but they should not be interpreted as a clean separation between mathematical content and surface form: dense embeddings can entangle semantic reasoning, lexical overlap, and formatting style.

*Table 12.* **Auxiliary Semantic Embedding Analysis.** We designate SLM-RFT as the reference anchor ($FD = 0, CS = 1$). The abbreviations are FD for Fréchet Distance, CS for Cosine Similarity, and SD for Semantic Diversity.

| Dataset | FD ($\downarrow$) | CS ($\uparrow$) | SD |
|---|---|---|---|
| **NuminaMath Subset** | 0.0184 | 0.9931 | 0.5363 |
| **SLM-RFT (Anchor)** | **0.0000** | **1.0000** | 0.5316 |
| **Oracle-Refined** | **0.0111** | **0.9955** | **0.5318** |
| **Oracle-Synthesized** | 0.1044 | 0.9361 | 0.4603 |

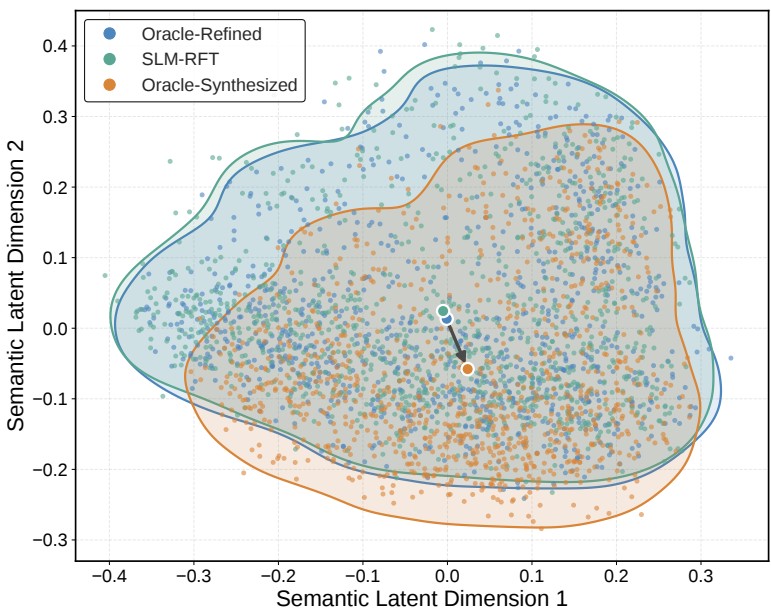

*Figure 6.* **Auxiliary Semantic Space Visualization (PCA).** KDE density contours provide a coarse view of embedding-space proximity, but the projection may entangle content, lexical overlap, and formatting style.

Table 12 and Figure 6 show that Oracle-Refined remains near the SLM-RFT anchor in embedding space, whereas Oracle-Synthesized is farther away. This supports the intuition that refinement starts from the native SLM trajectory rather than from a fully independent teacher trajectory. However, because the same embeddings can respond to both informational content and stylistic form, these results are not used as the primary evidence for semantic preservation. The main text therefore relies on the targeted judge evaluation in Table 3.

# E. Qualitative Case Studies of Style-Aligned Refinement

This section presents full, unedited generation trajectories to illustrate the *Quality-Utility Paradox*. By juxtaposing the raw outputs vertically, the contrast in syntactic density and structural formatting becomes visible. We focus on two mechanisms.

1. **Logic Repair with Procedural Preservation.** This case shows that Style-Aligned Refinement can repair logical errors while retaining the verbose trial-and-error reasoning steps that standard Oracle refinement compresses.

2. **Syntactic Preservation.** This case shows how the strategy preserves native scaffolding, such as enumerated lists, that Oracle models often flatten into dense paragraphs.

## E.1. Case Study 1 on Logic Repair with Style Preservation

---

**Problem Prompt**

Anna and Kati are celebrating their birthdays today. Three years from now, Anna will be four times as old as Kati was when Anna was two years older than Kati is now. How old is Anna if Kati is currently in high school?

---

**1. SLM-RFT**

Let Anna's current age be $a$ and Kati's current age be $k$.\n\nFrom the given information, we can write two equations:\n1. In three years, Anna will be $a + 3$ years old, and Kati will be $k + 3$ years old. At that time, Anna will be four times as old as Kati was when Anna was two years older than Kati is now. This can be written as:\n\\[\na + 3 = 4\\left(k - (a - 2 - k)\\right)\n\\]\n2. Since Kati is currently in high school, let's assume she is between 13 and 18 years old.\n\nSolving the first equation, we get:\n\\[\n**a + 3 = 4(2k - a) [Logic Error]**\n\\]\n\\[\na + 3 = 8k - 4a\n\\]\n\\[\n5a = 8k - 3\n\\]\n\nSince Kati is between 13 and 18 years old, we can test a few values of $k$ to find the corresponding value of $a$. For example, if $k = 13$, then:\n\\[\n5a = 8(13) - 3\n\\]\n\\[\n5a = 104 - 3\n\\]\n\\[\n5a = 101\n\\]\n\\[\na = \\frac{101}{5} = 20.2\n\\]\nNot a whole number, so invalid.\n\nTesting $k = 14$:\n\\[\n5a = 8(14) - 3 = 112 - 3 = 109\n\\]\n\\[\na = \\frac{109}{5} = 21.8\n\\]\nNot a whole number.\n\nTesting $k = 15$:\n\\[\n5a = 8(15) - 3 = 120 - 3 = 117\n\\]\n\\[\na = \\frac{117}{5} = 23.4\n\\]\nNot a whole number.\n\nTesting $k = 16$:\n\\[\n5a = 8(16) - 3 = 128 - 3 = 125\n\\]\n\\[\na = \\frac{125}{5} = 25\n\\]\nWhole number, valid.\n\nTherefore, Anna is $\\boxed{25}$.

---

**Analysis.** The generation explicitly details the trial-and-error attempts (Testing $k = 13, 14, \dots$). Despite the calculation error (highlighted in red), this verbose structure preserves the model's native reasoning style.

---

**2. Oracle-Refined**

Let Anna's current age be $a$ and Kati's current age be $k$.\n\nThree years from now, Anna will be $a+3$. We are told this equals four times Kati's age at an earlier time when "Anna was two years older than Kati is now," i.e. at a time $t$ such that Anna's age then was $k+2$ (since Anna is now $k$).\n\nThat time $t$ was $(a-(k+2))$ years ago, so Kati's age then was\n\\[\nk-\\bigl(a-(k+2)\\bigr)=2k-a+2.\n\\]\nThus the condition becomes\n\\[\na+3 = 4(2k-a+2).\n\\]\nSolve:\n\\[\na+3=8k-4a+8 \\quad\\Rightarrow\\quad 5a=8k+5 \\quad\\Rightarrow\\quad a=\\frac{8k+5}{5}.\n\\]\n\nSince Kati is currently in high school, take $k$ to be an integer from 13 to 18. We need $8k+5$ divisible by $5$, i.e. $8k\\equiv 0\\pmod 5$, so $k\\equiv 0\\pmod 5$. In the range 13–18, this gives $k=15$, hence\n\\[\na=\\frac{8(15)+5}{5}=\\frac{125}{5}=25.\n\\]\n\nTherefore, Anna is $\\boxed{25}$.

---

**Analysis.** The refined output adopts a compact presentation, consolidating multiple logical steps into single lines using composite symbolic operators such as `\Rightarrow`. Furthermore, it abstracts the trial-and-error process into a direct modular arithmetic constraint ($8k \equiv 0 \pmod 5$).

---

**3. Style-Aligned (GPT-5.2)**

```
Let Anna's current age be $a$ and Kati's current age be $k$.\n\n"In three years" Anna
will be $a+3$. This will equal four times Kati's age at the earlier time when "Anna
was two years older than Kati is now," meaning Anna's age then was $k+2$.\n\nSo that
earlier time was $a-(k+2)$ years ago, and Kati's age then was\n\\[\nk - \\bigl(a-(k+2)
\\bigr)=k-a+k+2=2k-a+2.\n\\]\nNow translate the statement:\n\\[\na+3 = 4(2k-a+2).\n\\]\
nSolve:\n\\[\na+3 = 8k-4a+8\n\\]\n\\[\n5a = 8k+5\n\\]\n\\[\na=\\frac{8k+5}{5}.\n\\]\n\
nSince Kati is in high school, take $k$ to be an integer from $13$ to $18$. For $a$ to
 be an integer, we need $8k+5$ divisible by $5$, i.e. $8k\\equiv 0 \\pmod 5$, so $k\\
equiv 0 \\pmod 5$. In that range, $k=15$ works.\n\nThen\n\\[\na=\\frac{8(15)+5}{5}=\\
frac{120+5}{5}=25.\n\\]\n\nSo Anna is \\boxed{25}.
```

**Analysis.** While maintaining the overall verbose structure, the model still condenses specific mathematical operations. For instance, it consolidates the algebraic expansion into a single line and identifies the correct value ($k = 15$) directly, omitting the unsuccessful trial attempts found in the native baseline.

---

**4. Style-Aligned (Qwen)**

```
Let Anna's current age be $a$ and Kati's current age be $k$.\n\nFrom the given
information, we can write two equations:\n1. In three years, Anna will be $a + 3$
years old, and Kati will be $k + 3$ years old. At that time, Anna will be four times
as old as Kati was when Anna was two years older than Kati is now. This can be written
 as:\n\\[\na + 3 = 4\\left(k - (a - 2 - k)\\right)\n\\]\n2. Since Kati is currently in
 high school, let's assume she is between 13 and 18 years old.\nSolving the first
equation, we get:\n\\[\na + 3 = 4(2k - a + 2)\n\\]\n\\[\na + 3 = 8k - 4a + 8\n\\]\n\\[\
n5a = 8k + 5\n\\]\n\nSince Kati is between 13 and 18 years old, we can test a few
values of $k$ to find the corresponding value of $a$. For example, if $k = 13$, then:\
n\\[\n5a = 8(13) + 5\n\\]\n\\[\n5a = 104 + 5\n\\]\n\\[\n5a = 109\n\\]\n\\[\na = \\frac
{109}{5} = 21.8\n\\]\nNot a whole number, so invalid.\n\nTesting $k = 14$:\n\\[\n5a =
8(14) + 5 = 112 + 5 = 117\n\\]\n\\[\na = \\frac{117}{5} = 23.4\n\\]\nNot a whole
number.\n\nTesting $k = 15$:\n\\[\n5a = 8(15) + 5 = 120 + 5 = 125\n\\]\n\\[\na = \\
frac{125}{5} = 25\n\\]\nWhole number, valid.\n\nTherefore, Anna is $\\boxed{25}$.
```

**Analysis.** The Qwen-based refinement mirrors the native baseline more closely. It reproduces the full trial-and-error sequence, including the explicit rejection of invalid hypotheses ($k = 13, 14$). This suggests that aligning within the same model family helps preserve the procedural details that are characteristic of the native distribution.

### E.2. Case Study 2 on Syntactic Structure Preservation

**Problem Prompt**

```
Given that the function $f(x)$ is an odd function defined on $\\mathbb{R}$, and when
$x \\leqslant 0$, $f(x)=-x^{2}-3x$, find $f(2)=$ \\_\\_\\_\\_\\_\\_\\_.
```

**1. SLM-RFT**

```
Let's solve the problem step by step.\n\n1. Since \\( f(x) \\) is an odd function, it
satisfies the property \\( f(-x) = -f(x) \\).\n2. When \\( x \\leq 0 \\), the function
```

```
   is given by \\( f(x) = -x^2 - 3x \\). We need to find \\( f(2) \\).\n3. To use the
property of the odd function, we will find \\( f(-2) \\) first, which we can then use
to find \\( f(2) \\).\n\nGiven \\( f(x) = -x^2 - 3x \\) for \\( x \\leq 0 \\), we have
:\n\\[\nf(-2) = -(-2)^2 - 3(-2) = -4 + 6 = 2.\n\\]\n\nSince \\( f(x) \\) is an odd
function, we know that \\( f(-2) = -f(2) \\). Therefore:\n\\[\n2 = -f(2) \\implies f(2)
 = -2.\n\\]\n\nThe value of \\( f(2) \\) is \\(\\boxed{-2}\\).
```

**Analysis.** The model uses a clear, numbered list structure (1., 2., 3.) to organize its reasoning steps, providing explicit visual separation between logical stages.

### 2. Oracle-Refined

```
Since \\(f(x)\\) is an odd function, it satisfies \\(f(-x)=-f(x)\\).\n\nFor \\(x\\le
0\\), \\(f(x)=-x^{2}-3x\\). Compute \\(f(-2)\\) (which is in the domain \\(x\\le 0\\))
:\n\\[\nf(-2)=-(-2)^2-3(-2)=-4+6=2.\n\\]\nUsing oddness, \\(f(-2)=-f(2)\\), so\n\\[\n2
=-f(2)\\implies f(2)=-2.\n\\]\n\nTherefore, \\(f(2)=-2\\).
```

**Analysis.** The refinement removes the numbered list structure, merging the reasoning steps into a dense paragraph. This change increases the information density by removing the visual delimiters used in the native reasoning trace.

### 3. Style-Aligned (GPT-5.2)

```
Let's solve the problem step by step.\n\n1. Since \\( f(x) \\) is an odd function, it
satisfies \\( f(-x) = -f(x) \\).\n2. When \\( x \\le 0 \\), we are given \\( f(x) = -x
^2 - 3x \\). To find \\( f(2) \\), it's easiest to compute \\( f(-2) \\) first (
because \\(-2 \\le 0\\)).\n3. Compute:\n\\[\nf(-2) = -(-2)^2 - 3(-2) = -4 + 6 = 2.\n
\\]\n4. Now use oddness: \\( f(-2) = -f(2) \\). So:\n\\[\n2 = -f(2)\\ \\Rightarrow\\ f
(2) = -2.\n\\]\n\n\\boxed{-2}
```

**Analysis.** Although the model restores the numbered list format, it introduces additional explanatory phrases, such as *because* $-2 \leq 0$, that were not present in the native baseline. This indicates a slight stylistic drift toward more didactic explanations.

### 4. Style-Aligned (Qwen)

```
Let's solve the problem step by step.\n\n1. Since \\( f(x) \\) is an odd function, it
satisfies the property \\( f(-x) = -f(x) \\).\n2. When \\( x \\leq 0 \\), the function
 is given by \\( f(x) = -x^2 - 3x \\). We need to find \\( f(2) \\).\n3. To use the
property of the odd function, we will find \\( f(-2) \\) first, which we can then use
to find \\( f(2) \\).\n\nGiven \\( f(x) = -x^2 - 3x \\) for \\( x \\leq 0 \\), we have
:\n\\[\nf(-2) = -(-2)^2 - 3(-2) = -4 + 6 = 2.\n\\]\n\nSince \\( f(x) \\) is an odd
function, we know that \\( f(-2) = -f(2) \\). Therefore:\n\\[\n2 = -f(2) \\implies f(2)
 = -2.\n\\]\n\nThe value of \\( f(2) \\) is \\(\\boxed{-2}\\).
```

**Analysis.** The output reproduces the native baseline's phrasing and list structure almost exactly, without adding extraneous explanations. Notably, despite lagging behind GPT-5.2 in general reasoning capabilities, the Qwen-based refiner captures the specific stylistic nuances of the source model more accurately.

# F. Detailed Token Frequency Statistics

This section provides a granular view of the vocabulary distribution shift discussed in Section 5.1. Table 13 lists the top 20 most frequent tokens for each dataset variant.

To visualize the structural divergence, we use color coding to highlight contrasting syntactic patterns. **Red rows** indicate *syntactic compaction*, a characteristic of Oracle-generated text where dense symbols (e.g., the raw backslash '\') displace explicit separators. **Blue rows** indicate *natural scaffolding*, reflecting the SLM's native preference for natural language connectives (e.g., ' the',' of') and spaced operators. The resurgence of these natural delimiters in the Style-Aligned datasets suggests partial restoration of the SLM-native distribution.

*Table 13.* **Top 20 Token Frequency Analysis.** Comparative vocabulary distribution. Oracle datasets prioritize compact syntax (Red), while native and Style-Aligned datasets favor natural delimiters (Blue). The **Style-Aligned (Qwen)** distribution closely mirrors the **SLM-RFT** baseline, supporting the role of distributional alignment.

### Oracle-Synthesized

| Rank | Token | Freq. | % |
|---|---|---|---|
| 1 | '\' | 1,062,630 | 7.39% |
| 2 | '2' | 602,712 | 4.19% |
| 3 | '1' | 553,789 | 3.85% |
| 4 | ' \(' | 426,189 | 2.97% |
| 5 | '0' | 385,283 | 2.68% |
| 6 | '3' | 284,719 | 1.98% |
| 7 | ' ' | 264,313 | 1.84% |
| 8 | '{' | 250,343 | 1.74% |
| 9 | ' \' | 242,945 | 1.69% |
| 10 | 'frac' | 197,830 | 1.38% |
| 11 | '.' | 197,028 | 1.37% |
| 12 | '[\n' | 194,757 | 1.35% |
| 13 | '=' | 193,836 | 1.35% |
| 14 | '4' | 192,128 | 1.34% |
| 15 | '-' | 191,259 | 1.33% |
| 16 | '\)' | 189,370 | 1.32% |
| 17 | '5' | 178,454 | 1.24% |
| 18 | '}{' | 144,724 | 1.01% |
| 19 | '+' | 134,439 | 0.94% |
| 20 | '^' | 133,963 | 0.93% |

### NuminaMath Subset

| Rank | Token | Freq. | % |
|---|---|---|---|
| 1 | ' ' | 460,262 | 4.09% |
| 2 | ' \' | 433,208 | 3.85% |
| 3 | '2' | 381,704 | 3.39% |
| 4 | '1' | 345,697 | 3.07% |
| 5 | ' the' | 333,719 | 2.96% |
| 6 | '0' | 252,217 | 2.24% |
| 7 | ' =' | 242,022 | 2.15% |
| 8 | ',' | 236,998 | 2.10% |
| 9 | '{' | 224,583 | 1.99% |
| 10 | ' $' | 196,654 | 1.75% |
| 11 | '3' | 180,476 | 1.60% |
| 12 | ' of' | 158,101 | 1.40% |
| 13 | '}' | 152,824 | 1.36% |
| 14 | '.' | 132,432 | 1.18% |
| 15 | '\' | 131,654 | 1.17% |
| 16 | '4' | 131,298 | 1.17% |
| 17 | ' +' | 129,443 | 1.15% |
| 18 | '5' | 126,428 | 1.12% |
| 19 | 'frac' | 121,964 | 1.08% |
| 20 | ' -' | 118,732 | 1.05% |

### Oracle-Refined

| Rank | Token | Freq. | % |
|---|---|---|---|
| 1 | '\' | 685,467 | 5.94% |
| 2 | '2' | 465,409 | 4.03% |
| 3 | '1' | 409,397 | 3.55% |
| 4 | '0' | 286,564 | 2.48% |
| 5 | ' ' | 277,254 | 2.40% |
| 6 | ' \(' | 276,592 | 2.40% |
| 7 | ' \' | 243,088 | 2.10% |
| 8 | '{' | 235,179 | 2.04% |
| 9 | ' the' | 223,106 | 1.93% |
| 10 | '3' | 210,389 | 1.82% |
| 11 | 'frac' | 160,320 | 1.39% |
| 12 | ',' | 157,962 | 1.37% |
| 13 | '4' | 150,702 | 1.30% |
| 14 | '[\n' | 137,401 | 1.19% |
| 15 | '5' | 136,539 | 1.18% |
| 16 | '}{' | 130,453 | 1.13% |
| 17 | '\)' | 126,402 | 1.09% |
| 18 | 'x' | 117,918 | 1.02% |
| 19 | ' =' | 114,422 | 0.99% |
| 20 | '\n' | 111,913 | 0.97% |

### Style-Aligned (GPT-5.2)

| Rank | Token | Freq. | % |
|---|---|---|---|
| 1 | '\' | 566,075 | 4.71% |
| 2 | '2' | 478,570 | 3.98% |
| 3 | ' ' | 477,056 | 3.97% |
| 4 | '1' | 414,804 | 3.45% |
| 5 | ' \' | 400,790 | 3.34% |
| 6 | '0' | 280,823 | 2.34% |
| 7 | '{' | 244,479 | 2.04% |
| 8 | ' the' | 243,760 | 2.03% |
| 9 | ' =' | 241,650 | 2.01% |
| 10 | ' \(' | 232,252 | 1.93% |
| 11 | '3' | 212,627 | 1.77% |
| 12 | ',' | 161,170 | 1.34% |
| 13 | 'frac' | 160,301 | 1.33% |
| 14 | '4' | 152,580 | 1.27% |
| 15 | '}{' | 136,460 | 1.14% |
| 16 | '5' | 136,373 | 1.14% |
| 17 | '[\n' | 133,074 | 1.11% |
| 18 | '}' | 129,009 | 1.07% |
| 19 | ' +' | 118,796 | 0.99% |
| 20 | '.' | 117,118 | 0.98% |

### Style-Aligned (Qwen)

| Rank | Token | Freq. | % |
|---|---|---|---|
| 1 | ' \' | 626,280 | 4.40% |
| 2 | ' ' | 602,608 | 4.24% |
| 3 | '2' | 492,715 | 3.47% |
| 4 | '1' | 426,858 | 3.00% |
| 5 | ' the' | 419,574 | 2.95% |
| 6 | ' =' | 364,393 | 2.56% |
| 7 | '\' | 342,222 | 2.41% |
| 8 | '0' | 289,593 | 2.04% |
| 9 | '{' | 287,027 | 2.02% |
| 10 | ',' | 247,946 | 1.74% |
| 11 | ' \(' | 228,104 | 1.60% |
| 12 | '3' | 227,359 | 1.60% |
| 13 | ' +' | 196,910 | 1.38% |
| 14 | ' -' | 192,405 | 1.35% |
| 15 | ' of' | 190,705 | 1.34% |
| 16 | '}' | 173,004 | 1.22% |
| 17 | ')' | 165,529 | 1.16% |
| 18 | 'frac' | 165,351 | 1.16% |
| 19 | '.' | 164,571 | 1.16% |
| 20 | '4' | 162,809 | 1.14% |

### SLM-RFT

| Rank | Token | Freq. | % |
|---|---|---|---|
| 1 | ' \' | 599,228 | 4.21% |
| 2 | ' ' | 594,487 | 4.17% |
| 3 | '2' | 485,146 | 3.41% |
| 4 | ' the' | 426,454 | 2.99% |
| 5 | '1' | 423,879 | 2.98% |
| 6 | ' =' | 361,138 | 2.54% |
| 7 | '\' | 336,914 | 2.37% |
| 8 | '0' | 289,445 | 2.03% |
| 9 | '{' | 281,540 | 1.98% |
| 10 | ',' | 251,963 | 1.77% |
| 11 | '3' | 225,655 | 1.58% |
| 12 | ' \(' | 213,835 | 1.50% |
| 13 | ' of' | 192,929 | 1.35% |
| 14 | ' +' | 191,580 | 1.35% |
| 15 | ' -' | 187,664 | 1.32% |
| 16 | '}' | 168,406 | 1.18% |
| 17 | '.' | 164,715 | 1.16% |
| 18 | 'frac' | 160,993 | 1.13% |
| 19 | '4' | 160,561 | 1.13% |
| 20 | ')' | 158,697 | 1.11% |

# G. Clarifying Apparent Discrepancies with Prior Work

This appendix discusses prior findings that may appear to conflict with the Quality-Utility Paradox. We focus on cases where stronger teachers or stronger generators are reported to produce better training data for smaller models. Our claim is not that strong-teacher data is inherently harmful. Rather, the paradox arises when Oracle-side logical improvement is coupled with a shift away from the target SLM's native reasoning distribution. When experimental settings suppress this distributional mismatch, or when teacher and student outputs are already stylistically compatible, stronger-teacher data can still be beneficial.

For example, OpenMathInstruct-2 reports that strong-teacher data can outperform equally sized weak-student data for LLaMA3.1-8B-scale training. The key difference is that its pipeline enforces a shared concise CoT format and applies rule-based post-processing to dense arithmetic expressions, reducing cross-source syntactic discrepancy. In contrast, our controlled setting fixes the problem set and directly compares SLM-native traces with Oracle-refined versions of the same traces, exposing the coupled effect of logical repair and Oracle-specific stylistic drift. Table 14 summarizes this distinction.

*Table 14.* **Reconciling Apparently Conflicting Strong-Teacher Findings.** The Quality-Utility Paradox arises when logical improvement is coupled with distributional drift away from the target SLM. Findings that favor stronger teachers are compatible with our claim when their settings reduce this drift or alter the optimization objective.

| Work | Reported Finding | Apparent Tension | Reconciliation |
|---|---|---|---|
| Toshniwal et al. (2024) | Section 2.2.2 reports that equally sized strong-teacher data can outperform weak-student data for LLaMA3.1-8B-scale training. | Our Oracle-Refined data has higher perceived quality but underperforms SLM-RFT data. | OpenMathInstruct-2 enforces a shared concise CoT format and applies rule-based post-processing to dense arithmetic expressions, reducing cross-source syntactic discrepancy. Logical quality can therefore dominate. |
| Bansal et al. (2025) | Their compute-matched results favor cheaper weaker generators, while some number-matched comparisons show stronger teachers outperforming weaker ones. | The stronger-teacher result may appear to conflict with our finding that a stronger Oracle can reduce utility. | Their setting mixes generator strength, sampling budget, and model-family effects. When stronger-teacher data is stylistically compatible with the learner, logical quality can dominate; our paradox targets the regime where Oracle refinement shifts data away from the learner's native distribution. |

# H. Scalability and Generalization Analysis

To examine whether the *Quality-Utility Paradox* is specific to a single model architecture or scale, we extend our evaluation to **Qwen2.5-Math-7B**, **LLaMA-3.2-3B**, and **DeepSeek-Math-7B**.

*Table 15.* **Scalability Analysis Across Model Scales and Training Paradigms.** We evaluate the downstream utility (Avg@16 Accuracy) of different data sources across four target models, comparing both Supervised Fine-Tuning (SFT) and Dynamic Fine-Tuning (DFT).

| Target Model | Dataset Source | Method | Downstream Utility (Avg@16 Accuracy) | | | | | Avg. |
|---|---|---|---|---|---|---|---|---|
| | | | MATH500 | AIME24 | AMC23 | Minerva | Olympiad | |
| **Qwen2.5-Math-1.5B** | Oracle-Synthesized | SFT | 51.6 | 0.0 | 35.0 | 13.6 | 16.1 | 23.26 |
| | | DFT | 64.8 | 3.3 | 35.0 | 19.1 | 27.9 | 30.02 |
| | NuminaMath Subset | SFT | 41.4 | 0.0 | 15.0 | 15.1 | 12.1 | 16.72 |
| | | DFT | 64.0 | 6.7 | 42.5 | 19.5 | 23.7 | 31.28 |
| | Oracle-Refined | SFT | 46.8 | 0.0 | 22.5 | 14.0 | 14.7 | 19.60 |
| | | DFT | 67.8 | 10.0 | 40.0 | 22.1 | 30.4 | 34.06 |
| | **SLM-RFT** | SFT | 49.8 | 0.0 | 35.0 | 13.2 | 15.7 | 22.74 |
| | | **DFT** | **70.0** | **10.0** | **40.0** | **32.0** | **33.3** | **37.06** |
| **Qwen2.5-Math-7B** | Oracle-Synthesized | SFT | 61.0 | 3.3 | 40.0 | 17.6 | 23.9 | 29.16 |
| | | DFT | 74.4 | 3.3 | 47.5 | 23.2 | 34.8 | 36.64 |
| | NuminaMath Subset | SFT | 53.4 | 6.7 | 27.5 | 22.8 | 19.4 | 25.96 |
| | | DFT | 71.2 | 6.7 | 42.5 | 29.0 | 32.6 | 36.40 |
| | Oracle-Refined | SFT | 58.4 | 10.0 | 37.5 | 16.2 | 21.5 | 28.72 |
| | | DFT | 73.8 | 6.7 | 57.5 | 27.2 | 33.5 | 39.74 |
| | **SLM-RFT** | SFT | 60.8 | 6.7 | 40.0 | 21.7 | 28.1 | 31.46 |
| | | **DFT** | 73.6 | 13.3 | 65.0 | 36.0 | 36.9 | 44.96 |
| **LLaMA-3.2-3B** | Oracle-Synthesized | SFT | 7.6 | 0 | 0 | 1.8 | 2.5 | 2.38 |
| | | DFT | 10.8 | 0 | 2.5 | 4 | 2.7 | 4 |
| | NuminaMath Subset | SFT | 5.8 | 0 | 2.5 | 3.3 | 2.2 | 2.76 |
| | | DFT | 11.4 | 0 | 2.7 | 5.9 | 2.2 | 4.44 |
| | **SLM-RFT** | SFT | 11.6 | 0 | 7.5 | 3.7 | 1.8 | 4.92 |
| | | **DFT** | 13.2 | 0 | 5.2 | 6.8 | 2.5 | 5.54 |
| **DeepSeek-Math-7B** | Oracle-Synthesized | SFT | 27.6 | 0 | 8.4 | 9.6 | 7.1 | 10.54 |
| | | DFT | 37.8 | 3.3 | 17.5 | 13.6 | 11.1 | 16.66 |
| | NuminaMath Subset | SFT | 26.5 | 0 | 7.8 | 8.2 | 5.9 | 9.68 |
| | | DFT | 40.4 | 3.3 | 15.7 | 17.6 | 15.3 | 18.46 |
| | **SLM-RFT** | SFT | 22.8 | 0 | 2.5 | 8.8 | 3.4 | 7.5 |
| | | **DFT** | 42.3 | 3.3 | 18.6 | 16.9 | 14.6 | 19.14 |

