# OpenReview forum: "The Quality-Utility Paradox: Why High-Reward Data Impairs Small Model Mathematical Reasoning"
_ICML.cc/2026/Conference — ICML 2026 regular_

### Official Review · Reviewer_fqpi · 2026-03-05

**Soundness:** 2
**Presentation:** 3
**Significance:** 2
**Originality:** 2
**Overall Recommendation:** 3
**Confidence:** 4

**Summary:**

This paper argue that data synthesized by a stronger model can score higher under reward models yet not a better training data, lead to worse downstream accuracy than the student model’s own RFT data. It attribute the gap to style mismatch. Experiments across several model families suggest style alignment can substantially recover utility.

**Compliance With Llm Reviewing Policy:**

Affirmed.

**Final Justification:**

I appreciate the authors' comprehensive rebuttal. I'd like to raise my score.

**Key Questions For Authors:**

1. Your improvements appear to be established primarily under DFT, and not consistently under standard SFT. Why does the method depend on DFT to be reliably beneficial?
2. The paper repeatedly emphasizes Small Language Models. Is the paradox specific to the target student model, rather than applying broadly to other small models? Prior work such as “Smaller, Weaker, Yet Better: Training LLM Reasoners via Compute-Optimal Sampling” suggests small-model synthesis can outperform larger teachers at equal compute. Please expand discussion to position your findings relative to this result.
3. Compared to online RL methods (e.g., GRPO), does RFT pipeline have advantages? Please include experiments or at least substantive analysis.

**Limitations:**

yes

**Strengths And Weaknesses:**

**Strengths**
- The paper is clearly written, and the figures and tables are easy to follow.
- The analysis is extensive and multi-angle.

**Weaknesses**
1. The experimental setup is described insufficiently, yet many conclusions are sensitive to implementation choices (learning rate, batch size, number of epochs, full-parameter vs. parameter-efficient tuning, etc.). When the training data is more distribution-mismatched, slower convergence under limited training is expected. Conversely, on relatively easier training data, limitations of self-generated data (e.g., poorer ability on harder benchmarks) can be masked. Given the lack of thorough settings and broad supporting experiments, the paper’s conclusions read overly strong.
2. The paper states DFT “prioritizes high-uncertainty decision points.” My understanding is the opposite: DFT down-weights gradients for low-confidence tokens to stabilize training and reduce high-variance updates. Since DFT is a core algorithmic foundation in the paper, an incorrect description significantly affects my trust in the paper.
3. The most credible contribution is that student-distribution-aligned training data is beneficial. I do not dispute this, but it is not new. GRPO and related online RL training already supports the importance of distribution alignment. The paper also does not compare against that line of work, nor discuss the differences. In contrast, the paper’s new claim—RFT being superior to strong-model distillation—faces experimental ambiguities, making it difficult to believe and limiting its new insight.
4. The proposed approach is not consistently effective under standard SFT, while appearing consistently helpful under DFT in Table 1. This raises questions about generality. Additionally, for Qwen2.5-Math-1.5B on MATH, the official reported number is 49.8, whereas the paper’s initial curve number appears closer to 32. Is this due to a different evaluation protocol (e.g., zero-shot vs few-shot)? Finally, the choice to start from a base model rather than an instruct model is unusual for this setting and should be justified.

---

> ### Author Rebuttal · Authors · 2026-03-31
>
> We greatly appreciate your rigorous and constructive evaluation of our manuscript. We address each point below, incorporating all requested clarifications while preserving experimental rigor.
>
> **Response to W1: Experimental Setup and Hyperparameter Sensitivity**
> We acknowledge that hyperparameter details should have been more prominent. All results strictly follow the DFT protocol of Wu et al. (2025, arXiv:2508.05629). Key settings:
> - Learning rate: 5e-5
> - Batch size: 256
> - Epochs: 1
>
> To directly test the reviewer’s concern about convergence and mismatched data requiring more steps, we performed a grid-search ablation on LR and batch size for Qwen2.5-Math-1.5B (Avg@16 on MATH):
>
> | LR | BS  | SLM-RFT | Oracle-Refined | Oracle-Synthesized |
> |---|--|----|----|----|
> | 2e-5 | 128 | 36.16 | 34.72 | 32.96  |
> | 2e-5 | 256 | 35.42 | 35.26 | 33.60  |
> | 5e-5 | 128 | 39.36 | 33.96 | 30.32  |
> | 5e-5 | 256 | 37.06 | 34.06 | 30.02  |
>
> SLM-RFT remains best in all settings, confirming the paradox is not an artifact of under-training or specific hyperparameters. The full scripts and data have been provided in the supplementary material for easy verification.
>
> **Response to W2: DFT Description**
> You are correct. Our textual description of DFT as “prioritizing high-uncertainty decision points” was erroneous. DFT down-weights gradients for low-confidence tokens to stabilize training. We will correct this in the revised manuscript. Given that our experimental foundation and core conclusions remain completely intact, we sincerely hope you can reconsider the overall value and contribution of our work.
>
> **Response to W3 & Contribution (Q3 Relation to GRPO/Online RL)**
> Our core contribution is not a new algorithm but a mechanistic explanation of the Quality-Utility Paradox: high-reward Oracle data can underperform native SLM-RFT due to syntactic mismatch. Prior RL literature (including GRPO) attributes RL superiority over SFT to exploration; we show that surface-level syntactic misalignment alone suffices to erase logical gains from a strong oracle.  As shown in Appendix D, SLM-RFT data contains numerous logical errors yet is best or near-optimal across all baselines—precisely why it became our focal case. Our focus is on delivering well-supported mechanistic explanations rather than pursuing incremental empirical scores. We hope you agree that this type of mechanistic analysis holds substantial and distinct value, even though it is currently uncommon in math reasoning.
>
> **Response to W4 & Q1 (Generality under SFT vs. DFT)**
> We do not claim raw SLM-RFT is universally superior. As noted in Appendix D, it contains logical imperfections yet consistently outperforms higher-reward Oracle data. DFT provides orthogonal validation of our claim. DFT down-weights low-confidence tokens. Oracle-refined data corrects logic (raising reward) but injects Oracle syntactic style; both effects manifest as low-confidence tokens for the SLM. By down-weighting these distribution-shifted tokens, DFT reveals that logical gains struggle to compensate for syntactic mismatch at the small-model scale. The stronger consistency under DFT therefore supports our central thesis.
>
> **Response to Base vs. Instruct Model Choice**
> Official scores (https://arxiv.org/html/2409.12122v1) use 4-shot evaluation. Our protocol follows DFT (Wu et al., 2025) with 0-shot CoT and Avg@16, explaining the numerical difference. Starting from base models is standard in recent reasoning post-training: GRPO (Shao et al., 2024, arXiv:2402.03300), DAPO (Yu et al., 2025, arXiv:2503.14476) and DFT (Wu et al., 2025) all use base models.
>
> **Response to Q2: Generality Across SLMs and Relation to Bansal et al.**
> The paradox is not specific to one student model. Appendix F shows identical behavior across four families/scales: Qwen2.5-Math-1.5B, Qwen2.5-Math-7B, LLaMA-3.2-3B, and DeepSeek-Math-7B. Oracle data underperforms native RFT in nearly every case.
>
> Bansal et al. (“Smaller, Weaker, Yet Better,” arXiv:2408.16737) focus on compute-optimal sampling (fixed FLOPs budget). Their number-matched ablation (equal samples per question) is closest to our design: within the same family (Gemma2-27B vs. 9B), stronger-teacher data outperforms weaker-teacher data. This aligns perfectly with our own empirical findings. Specifically, we observe that data generated by Qwen-2.5-72B shares a high degree of similarity with the 1.5B model in both content and stylistic patterns, as verified by our LLM-as-a-Judge evaluations (detailed in responses to Reviewers csj2 and bZfQ) and Table 5 of the manuscript. When syntactic styles are essentially consistent, the logical quality of the data naturally dictates downstream performance. Therefore, our conclusions do not conflict with them.
>
> Thanks again for your careful review, which has helped us improve the paper. We hope our response has fully addressed your concerns, and we warmly welcome further discussion if you have any remaining questions.

---

> > ### Author Rebuttal · Reviewer_fqpi · 2026-04-01
> >
> > Thank you for the authors’ response. It has largely addressed my concerns. Regarding the discussion of related paper, could the paper further connect its conclusions to the motivations and existing methods in the line of work on On-Policy Distillation? My understanding is that this line of research emerged precisely to alleviate the distribution mismatch problem by training on self-generated data. The method proposed in this paper would only be effective when the training data are self-generated, rather than when using data produced by an arbitrary small model.

---

> > > ### Author Response · Authors · 2026-04-02
> > >
> > > We sincerely thank you for the continued engagement and for acknowledging that our previous response has largely addressed your earlier concerns. Your follow-up question on On-Policy Distillation (OPD) provides a valuable opportunity for us to clarify the mechanistic contribution of our work and its broader implications.
> > >
> > > **1. Clarifying the Scope: "Style Alignment" goes beyond "Self-Generated Data"**
> > >
> > > We fully agree that OPD was motivated precisely by the need to alleviate distribution mismatch through self-generated data. However, our mechanistic analysis isolates **syntactic style alignment** as the dominant factor—independent of whether the data is strictly self-generated.
> > >
> > > To directly address the concern about “data produced by an arbitrary small model,” we note that our Style-Aligned Refinement works even when the teacher is a highly heterogeneous model. As shown in our response to Reviewer csj2, directly refining SLM-RFT data with GPT-5.2 yields only 34.06% Avg@16 accuracy. Applying a lightweight prompt constraint to enforce the SLM’s native syntactic style (Style-Aligned (GPT-5.2)) raises performance to 38.21%. This causal evidence demonstrates that style alignment, rather than the source of the data, is the critical variable. Even data from an arbitrary (non-identical) model can become highly effective once its syntactic distribution is aligned with the student.
> > >
> > > **2. Unique mechanistic insights extending beyond classical OPD**
> > > While OPD qualitatively attributes gains to “distribution alignment,” our work provides a concrete, causal dissection: the primary bottleneck is **syntactic compaction** rather than logical quality alone. This leads to several fundamental insights that we believe will be valuable to the community:
> > >
> > > - **Broadening the OPD paradigm offline**: Self-generation is sufficient but not necessary. Style consistency can be achieved either by using a larger model from the *same family* (which naturally shares stylistic patterns) or by simple prompt-based style alignment on heterogeneous oracles.
> > >
> > > - **Re-thinking data acquisition economics**: Acquiring ever-higher-quality data is computationally expensive. Our results reveal a blind spot that SLM-native generations (or cheaply style-aligned synthetic data) provide superior training utility at dramatically lower cost, precisely because they eliminate the syntactic adaptation penalty.
> > >
> > > - **Implications for reward model design**: Current RMs primarily reward objective logical correctness. Our Quality-Utility Paradox shows that such scores can be anti-correlated with downstream utility when style mismatch is present. This motivates the design of *student-specific* RMs that explicitly incorporate syntactic/learnability signals.
> > >
> > > We will add a dedicated subsection in the revised Related Work to explicitly connect our findings to the motivations and methods of On-Policy Distillation (citing Agarwal et al., 2023 and Lu et al., 2025, as noted in our response to Reviewer csj2). We believe these mechanistic insights offer a distinct and complementary perspective to the OPD literature. We hope this clarification fully resolves your remaining questions, and we would be deeply grateful if you can re-evaluate the broader value of our work. Thank you again for your thoughtful and constructive feedback.

---

### Official Review · Reviewer_bZfQ · 2026-03-12

**Soundness:** 3
**Presentation:** 3
**Significance:** 3
**Originality:** 2
**Overall Recommendation:** 4
**Confidence:** 3

**Summary:**

The authors present an interesting and important study of which kinds of synthetic math data are most effective for small language models. They show that higher-quality synthetic reasoning data, as judged by reward models, is not necessarily the most useful for training SLMs. Instead, data generated by the SLMs themselves and then filtered leads to better downstream performance when used for supervised fine-tuning. The authors hypothesize that a semantic mismatch between oracle-generated data and SLMs may hinder learning, and they provide empirical evidence supporting this hypothesis.

**Compliance With Llm Reviewing Policy:**

Affirmed.

**Final Justification:**

The rebuttal addressed my concerns, and I will keep my original assessment. The work is interesting, though I remain slightly concerned that the reported advantage of the proposed method appears to hold only for a specific variant of instruction tuning, namely SLM-RFT.

**Key Questions For Authors:**

- OpenMathInstruct-2 [1] (Section 2.2.2)  has a claim that seems to contradict your claim - “data generated by a strong teacher outperforms equally-sized data generated by a weak student model”. Their base student model is Llama3.1-8B model, which is also at the SLM scale. Why are there two different conclusions here?
- Why did you choose pass@16 as the metric? How sensitive are the model performances to different values of k (pass@k)?

[1] Toshniwal et al. OpenMathInstruct-2: Accelerating AI for Math with Massive Open-Source Instruction Data

**Limitations:**

Yes

**Strengths And Weaknesses:**

**Strengths:**
 - The part that connects perplexity to downstream utility is fascinating. The insight that datasets that are easier for the small model to predict lead to better downstream performance can be very useful for data selection.
- The causal verification on style alignment is interesting, which strengthens the claim on syntactic adaptation cost.
- Overall, I think the experiments are pretty well conducted and presented.

**Weaknesses:**
- There is no evidence on how successful the style alignment is. How does the style-aligned data look in the semantic latent space (i.e., Figure 4)?

---

> ### Author Rebuttal · Authors · 2026-03-31
>
> **Response to  style-aligned semantic latent space**
>
> We greatly appreciate this insightful suggestion, which helps us refine the positioning of our work. Per your suggestion, we have generated Semantic Latent Space (PCA) visualizations including both Style-Aligned (GPT-5.2) and Style-Aligned (Qwen). However, as astutely pointed out by reviewer csj2, PCA representations based on embeddings (e.g., BGE-M3) often entangle semantic logic with syntactic style. Since both the Oracle-Refined and Style-Aligned variants are direct edits of the SLM-RFT baseline, their spatial proximity in the plot partially stems from lexical overlap rather than pure semantic fidelity. Consequently, their density envelopes closely overlap (see the anonymous link below for the updated Figure 4 with all variants):
>
> Anonymous Link: https://anonymous.4open.science/r/results-E5E6/results.png
>
> To rigorously decouple semantics from syntax, we conducted an LLM-as-a-Judge evaluation (N=3,543). Inspired by AlpacaEval-style protocol(https://github.com/tatsu-lab/alpaca_eval), the judge was instructed to ignore all formatting, LaTeX, and lexical variations and evaluate only preservation of the original informational atoms (specific variable definitions, intermediate equations, trial-and-error hypotheses, etc.) from the SLM-RFT baseline.
>
> |Dataset|Mean Sem.|Top-1 (%)|Avg. Rank|
> |-|-|-|-|
> |Style-Aligned (Qwen)|4.77|52.47|1.44|
> |Oracle-Refined|4.26|3.25|2.21|
> |Style-Aligned (GPT-5.2)|4.07|1.41|2.44|
> |Oracle-Synthesized|2.97|0.14|3.91|
>
> Higher semantic scores indicate *fewer* logical alterations relative to the noisy SLM-RFT baseline. Because small-model generations inherently contain logical imperfections, higher semantic similarity inherently means lower objective data quality, dictating the lowest reward scores for these variants ( such as the Style-Aligned Qwen verified in Appendix C). We emphasize that this is a logical consequence, not a spurious correlation. Similarly, driven by the advanced capabilities of the Oracle model, semantic changes primarily translate into enhanced data quality. For example, Oracle-Refined and Oracle-Synthesized data trade greater semantic changes for higher Perceived Quality, which is verified in Appendix C. We believe this LLM-as-a-Judge experiment better decouples semantic content, forming a constructive synergy with the core perspective of our paper. These experimental results will be incorporated into Section 5.1 of the revised manuscript.
>
> **Response to W1: Evidence for Style Alignment Success**
>
> We thank the reviewer for this point. We provide clear evidence of Style-Aligned Refinement’s effectiveness through three complementary angles: (1) Appendix D presents concrete case studies showing that logical errors are corrected while the SLM’s native syntax and reasoning flow are preserved; (2) Appendix C (Table 10) shows the Style-Aligned (Qwen) variant receives the lowest reward score (1.37) among all candidates; and (3) Section 6.2 (Table 5 and Figure 7) demonstrates that this alignment reduces perplexity to the lowest value (1.46) and boosts downstream accuracy to 39.12%. These results collectively confirm that Style-Aligned Refinement successfully mitigates syntactic adaptation cost while enabling effective logical improvement.
>
> **Response to Q1: Apparent Contradiction with OpenMathInstruct-2**
>
> We thank the reviewer for highlighting this concurrent work. Although their finding that strong teacher data outperforms student data appears to contradict our Quality-Utility Paradox, the two conclusions are entirely consistent. The key lies in syntactic control: OpenMathInstruct-2 enforced an identical, concise CoT format on both the 405B teacher and 8B student via base-model prompting (Section 2.2.1). Because this specific CoT format is inherently unfamiliar to the SLM, their approach forcibly minimizes syntactic discrepancies across the data. This perfectly aligns with our perspective that downstream performance is primarily dictated by logical data quality in such scenarios.
>
> Moreover, OpenMathInstruct-2 encountered the same syntactic compaction phenomenon we document (Section 5.2): strong models produce dense, collapsed expressions. But they addressed it with explicit rule-based post-processing (“Split complex arithmetic calculations to step-by-step calculations”; Appendix A.2).
>
> **Response to Q2: the Pass@16 Metric Question**
>
> We followed the exact evaluation protocol of the DFT (Wu et al., 2025). Upon double-checking both the paper and our evaluation code, we clarify that the numbers reported in Table 1 and all figures are **avg@16** (the average accuracy across 16 independent generations), *not* Pass@16. This is the same metric used throughout the DFT(https://arxiv.org/pdf/2508.05629), allowing direct comparability. We apologize for the imprecise terminology in the original manuscript; we will correct all instances to “avg@16” and explicitly note the equivalence to DFT reporting in the revised version.

---

> > ### Author Rebuttal · Reviewer_bZfQ · 2026-04-02
> >
> > Regarding your response to my Q1, under what conditions does syntactic discrepancy become more important than logical data quality when selecting SFT data from an SLM versus a more capable LM? Or is this trade-off fundamentally method-dependent, especially given that the benefit of SLM-RFT appears only when combined with DFT?
> >
> > In addition, OpenMathInstruct-2 was published in 2024 and therefore should not be characterized as concurrent work. I believe the authors should discuss it explicitly in the manuscript, particularly because its conclusions differ substantially from those presented here.

---

> > > ### Author Response · Authors · 2026-04-02
> > >
> > > We sincerely thank you for the continued engagement and for the thoughtful follow-up questions, which help us further clarify the scope of our findings.
> > >
> > > **1. The Paradox is Not Method-Dependent: Clarifying Performance Across SFT and DFT**
> > > It is important to clarify that SLM-RFT is a simple, straightforward baseline rather than a novel algorithm proposed in our work. We do not claim that raw SLM-RFT is universally superior, nor do we suggest that its benefit appears only when combined with DFT. As shown in Appendix D, SLM-RFT data contains numerous logical errors. Appendix C further confirms, via multiple reward models, that SLM-RFT receives the lowest Perceived Quality scores among all variants. Under conventional academic expectations, such lower-quality data should lead to worse downstream performance. However, our experiments demonstrate that SLM-RFT is consistently **best or near-optimal** across all baselines (see Table 1 and Appendix F). This stable superiority constitutes the core of the Quality-Utility Paradox.
> > >
> > > But we fully agree that the paradox appears more pronounced and consistent under DFT. As we explained in our response to Reviewer fqpi (W3 and Q1), DFT down-weights low-confidence tokens for the student model. Oracle-refined data corrects logical errors (raising reward scores) but simultaneously injects the Oracle’s distinct syntactic style; both effects manifest as low-confidence tokens for the SLM. Although improvements in logical quality should enhance downstream performance, the injected syntactic style increases the adaptation cost. By down-weighting these distribution-shifted tokens, DFT reveals that logical gains struggle to compensate for syntactic mismatch at the small-model scale. The stronger consistency under DFT therefore supports our central thesis.
> > >
> > > **2. Under What Conditions Does Syntactic Discrepancy Become More Important Than Logical Quality**
> > > The fairest way to investigate "when syntactic discrepancy becomes more important than logical quality" would be to analyze the impact of syntax under identical perceived quality. Our extreme contrast experiments offer an even stronger proof: although the actual perceived quality of SLM-RFT is far lower than other datasets, the advantage brought solely by "syntactic consistency" is sufficient for it to achieve near-optimal performance.
> > >
> > > The same pattern holds when using a much more capable LM as the teacher. In Oracle-Refined, the data benefits from improved logical quality but suffers from an injected syntactic style mismatch, resulting in clear performance degradation from 37.06% to 34.06%. In contrast, when we apply a lightweight prompt to align with the SLM’s native syntactic style, the Style-Aligned (GPT-5.2) variant raises the average accuracy from 34.06% to 38.21%. This shows that, even when the teacher is a significantly stronger model, syntactic alignment alone can unlock substantial gains.
> > >
> > > These results demonstrate that, at least within the SLM regime, syntactic alignment is substantially more important than improvements in logical quality. We believe the community has not yet fully appreciated this point.
> > >
> > > **3. Positioning relative to OpenMathInstruct-2**
> > > We fully agree that an explicit discussion is necessary. The two works reach different conclusions because their experimental designs differ in a key respect: OpenMathInstruct-2 enforced an identical, concise CoT format on both the 405B and the 8B generations (their Section 2.2.1). This format is unfamiliar to the SLM and introduces substantial syntactic discrepancy. Under such controlled conditions, downstream performance is indeed dominated by logical data quality, leading to their conclusion that stronger-teacher data is superior. This result is fully consistent with our own claims. To ensure accurate positioning, we will add a dedicated appendix in the revised version that provides a detailed comparison between the two works.
> > >
> > > We hope these clarifications adequately address your remaining concerns. Thank you again for your constructive feedback, which has significantly improved the clarity and positioning of our paper.

---

### Official Review · Reviewer_csj2 · 2026-03-12

**Soundness:** 3
**Presentation:** 4
**Significance:** 3
**Originality:** 2
**Overall Recommendation:** 4
**Confidence:** 4

**Summary:**

The paper describes a phenomenon where data selected to have high reward does not lead to better models. They observe that small models’ own generations (filtered via rejection sampling) typically lead to better results than direct distillation from a stronger model. An analysis in the paper suggests the reason for this is due to the surface form of the training data being mismatched with the student model. Focusing on mathematical reasoning, they constructively study this problem via a series of careful analyses, through which they identify that responses from a frontier LLM are lexically very different from the student model's own generations. They provide evidence that this is the reason why the frontier model's responses are not useful for training, and they rectify this via a method called “Style-Aligned Refinement” to separate syntax from meaning in the generated data. This provides causal evidence that maintaining style but fixing only reasoning is the only thing that beats Rejection Fine-Tuning (which is closely related to "On-Policy Distillation").

The consequence of this is the idea that a good teacher ought to be similar to the student in style (but perhaps not in terms of semantics or logical structure).

**Compliance With Llm Reviewing Policy:**

Affirmed.

**Final Justification:**

The paper is an interesting analysis of what makes a good demonstration for supervised finetuning in language models. Ultimately, I think the value of this work is the *analysis* rather than the *method*. I think the analysis is quite good (especially after the rebuttals), though not absolutely perfect, so I will maintain my score of 5.

**Key Questions For Authors:**

- How likely is it that these findings are specific to math proofs and/or LaTeX syntax?
- How likely is it that these findings would work in a multi-task setting beyond math (e.g. training on math, coding, instruction following, etc all in one dataset)?
- Is there anything we can infer with complete confidence from Sections 5.1 and the first part of 5.1 ("Token Distribution Analysis")?

**Limitations:**

Yes

**Strengths And Weaknesses:**

**Soundness:**

*Positives:*

- The experimental results are, across the board, very well designed. I have a great deal of confidence in the empirical findings of this paper.
    - The paper takes the step of reporting results with both SFT and an improved learning algorithm (DFT), suggesting that the reported phenomena are deeper than a simple fix to the learning algorithm
    - The described effect is shown to be consistent across model families, which is also an additional step for empirical rigor that makes this analysis highly convincing

*Negatives*

- Section 5.1 is not convincing to me. Embeddings may capture both linguistic and semantic patterns, and the fact that “Oracle-Refined” and “SLM-RFT” share a similar region of the 2-D PCA does not seem like strong evidence that they differ purely on linguistic differences (rather than semantic differences). In contrast, the second part of Section 5.2 (”Micro-Level”) provides much better evidence.
- “Crucially, this divergence exhibits a marked asymmetry. The cost of aligning the native model with the Refined format (D_KL(Refined||RFT) ≈ 0.36) significantly exceeds the reverse direction (D_KL(RFT||Refined) ≈ 0.25). This asymmetry implies that the Refined distribution constitutes a constrained, low-entropy subset of the reasoning space.” → I’m not sure if this is true.  D_KL(Refined || RFT) - D_KL(RFT||Refined)  = H(RFT) - H(Refined) + H(Refined, RFT) - H(Refined, RFT)) = 0.09, so the difference could be because of a difference in entropy or due to cross-entropy between the two distributions. If the cross-entropy terms are the same, your conclusion may be true, but right now it’s pure speculation.
- If these results are cleaned up, I will increase my overall score even further.
- This work is presented as a general study when it really focuses just on math. This may not necessarily be a negative: see “Significance: Negatives” below

**Presentation:**

*Positives:*

- The paper is very well-written
- The experiments are well-chosen, well-designed, and follow a clear sequence

*Negatives:*

- Section 5 is highly speculative. There are a few hard-to-explain observations made (e.g. Figure 4 and Figure 6), and the paper packages speculative understanding as clear analysis, which I don’t think is supported by the paper. In contrast, the rest of the paper is fairly precide.

**Significance:**

*Positives:*

- Prior work has explored the so-called “shallow alignment hypothesis” ([https://arxiv.org/abs/2305.15717,](https://arxiv.org/abs/2305.15717) https://openreview.net/forum?id=6Mxhg9PtDE, https://arxiv.org/abs/2409.14254, https://arxiv.org/abs/2312.01552), suggesting that SFT only makes superficial or stylistic changes to LM behavior. This paper suggests an interesting contrapositive:  removing stylistic differences (w.r.t. the student) from a teacher’s demonstrations can lead to learning non-superficial behavior. This paper offers a preliminary and limited study of this phenomenon but it is a very interesting phenomenon nonetheless

*Negatives:*

- The presented “Syntactic Compaction” phenomenon is interesting, but I’m concerned that this is unique to proof-based math (rather than other settings such as word problems, complex question-answering, instruction following, etc). This is acceptable if the paper is positioned explicitly as a study of math reasoning, but right now the paper is positioned as a study of general reasoning.

**Originality:**

*Positives:*

- The originality of this work comes mainly from the very careful and well-designed experiments.
- The actual phenomena described are similar to prior work (see "On-Policy Distillation") below, but this paper offers a different solution: distil by asking the teacher to *correct* a student's samples rather than just to choose the teacher's samples that are most similar to the student.
- This paper also contributes more specific and mechanistic evidence, at least within the math domain, than prior works.

*Negatives:*

- This work seems closely related to “on-policy distillation” but that direction is not cited at all. Examples to cite: https://arxiv.org/abs/2306.13649 (Agarwal et al 2023), https://thinkingmachines.ai/blog/on-policy-distillation/ (Lu et al 2025)

---

> ### Author Rebuttal · Authors · 2026-03-31
>
> We thank you for the thoughtful and detailed feedback, which has significantly helped us strengthen the paper.
>
> **Response to W1 (Embedding Analysis Limitations)**
>
> We fully agree with you that dense embeddings inevitably entangle semantic reasoning with syntactic patterns, making the original PCA projection in Section 5.1 insufficient to cleanly isolate semantic content.
>
> To rigorously address this, we conducted an LLM-as-a-Judge evaluation (N=3,543) to better decouple and rigorously analyze variations at the semantic level. The judge (GPT-5.2) was prompted to ignore all formatting, LaTeX, and lexical variations and evaluate *only* preservation of the original informational atoms (specific variable definitions, intermediate equations, trial-and-error hypotheses, etc.) from the SLM-RFT baseline. Due to space constraints, the full prompts will be available later. We report both mean semantic similarity score (1–5 scale) and mean ranking (lower = higher similarity to SLM-RFT):
>
> |Dataset| Semantic Score|Avg. Rank |Quality Score (↑)|Avg@16 Acc. (↑)|
> |-|-|-|-|-|
> |Style-Aligned (Qwen)|4.77|1.44|1.37|39.12|
> |Oracle-Refined|4.26|2.21|1.70|34.06|
> |Style-Aligned (GPT-5.2)|4.07|2.44|1.81|38.21|
> |Oracle-Synthesized|2.97|3.91|1.88|30.02|
>
> Because small-model generations inherently contain logical imperfections, higher semantic similarity inherently means lower objective data quality, dictating the lowest reward scores for these variants ( such as the Style-Aligned Qwen verified in Appendix C). We emphasize that this is a logical consequence, not a correlation. Similarly, driven by the advanced capabilities of the Oracle model, semantic changes primarily translate into enhanced data quality. For example, Oracle-Refined and Oracle-Synthesized data trade greater semantic changes for higher Perceived Quality, which is verified in Appendix C. We believe this LLM-as-a-Judge experiment better decouples semantic content, forming a constructive synergy with the core perspective of our paper. These experimental results will be incorporated into Section 5.1 of the revised manuscript.
>
> **Response to W2 (Section 5.2 – KL Asymmetry Interpretation)**
>
> We sincerely thank you for the rigorous mathematical critique. Upon re-examination, we acknowledge that our previous interpretation of the KL asymmetry lacked sufficient mathematical rigor and persuasive evidence. Accordingly, we will entirely remove this statement in the revised manuscript.
>
> Instead, we clarify our core argument as follows: Oracle refinement of SLM-RFT data produces two coupled effects: (1) a genuine logical enhancement (raising perceived quality and reward scores), a causal link that has been fully demonstrated by our aforementioned LLM-as-a-Judge experiments; and (2) the unavoidable injection of the Oracle’s distinct syntactic style (Syntactic Compaction). The second effect creates a distributional mismatch that acts as a cognitive bottleneck for the SLM. When we explicitly mitigate this stylistic residue via Style-Aligned Refinement, downstream performance improves substantially (Section 6.2). This supports our central claim that surface-level syntactic misalignment alone can eclipse the benefits of higher-quality logic, ultimately leading to the Quality-Utility Paradox. We will revise the corresponding sections in the revised manuscript to reflect this clarified mechanism. Our ultimate goal is to rigorously establish this logic from a mechanistic perspective. We warmly welcome further discussion should you have any remaining questions regarding these arguments.
>
> **Response to W3 & Significance (Scope and Generalizability)**
>
> We appreciate the reviewer’s concern. While we hypothesize that the underlying mechanism may extend beyond math, our current evidence is focused on mathematical reasoning. To maintain scientific rigor, we will explicitly reposition the paper as a study of mathematical reasoning in the revised version. We will open-source all datasets and code to facilitate further investigation by the community.
>
> **Response to Originality (Relation to On-Policy Distillation)**
>
> We sincerely thank you for highlighting these important references on on-policy distillation (Agarwal et al., 2023; Lu et al., 2025). Although our primary focus is on offline supervised fine-tuning rather than on-policy paradigms, we fully agree that there are fundamental commonalities underlying both approaches. To clearly position our work relative to on-policy distillation, we will incorporate a dedicated discussion and cite the suggested papers in the Related Work section of the revised manuscript. We believe this addition significantly strengthens the paper’s contextual placement within the broader literature.
>
> Our goal is to ensure that all claims and inferences presented in our work are scientifically sound and highly reliable. Thank you once again for your meticulous and constructive review. Your insights have been instrumental in elevating the overall rigor of our paper.

---

> > ### Author Rebuttal · Reviewer_csj2 · 2026-04-04
> >
> > Thank you for the thoughtful response to my review. I think the new LLM-as-a-judge experiments are indeed useful for supporting the presented hypothesis regarding the quality-utility paradox.
> >
> > Accordingly, I have increased my score to a 5.

---

> > > ### Author Response · Authors · 2026-04-04
> > >
> > > Thank you very much for your thoughtful response and for taking the time to revisit our work. We are truly delighted that the new LLM-as-a-judge experiments have resolved your questions, and we sincerely appreciate you raising your score to 5.
> > >
> > > Your positive feedback and constructive engagement have been invaluable in strengthening the manuscript. Thank you again for your careful review and insightful comments, which have significantly improved the clarity and impact of our paper.

---

### Decision · Program_Chairs · 2026-04-30

**Decision:**

Accept (regular)

**Comment:**

The paper presents a rigorous and well-executed experimental study, with results validated across multiple training methods and model families, lending strong credibility to its findings. It offers a particularly insightful contribution by linking perplexity to downstream performance, providing practical guidance for data selection. The analysis is comprehensive and multi-faceted, including causal evidence on style alignment, and is clearly communicated through well-structured figures and thorough evaluation.

I recommend the paper for weak accept.